# m6A RNA methylation impacts fate choices during skin morphogenesis

Linghe Xi[1], Thomas Carroll[2†], Irina Matos[1†], Ji-Dung Luo[2], Lisa Polak[1], H Amalia Pasolli[3], Samie R Jaffrey[4], Elaine Fuchs[1]*

[1]Howard Hughes Medical Institute, Robin Chemers Neustein Laboratory of Mammalian Cell Biology and Development, The Rockefeller University, New York, United States; [2]Bioinformatics Resource Center, The Rockefeller University, New York, United States; [3]Electron Microscopy Resource Center, The Rockefeller University, New York, United States; [4]Department of Pharmacology, Weill Cornell Medicine, Cornell University, New York, United States

**Abstract** $N^6$-methyladenosine is the most prominent RNA modification in mammals. Here, we study mouse skin embryogenesis to tackle m6A's functions and physiological importance. We first landscape the m6A modifications on skin epithelial progenitor mRNAs. Contrasting with in vivo ribosomal profiling, we unearth a correlation between m6A modification in coding sequences and enhanced translation, particularly of key morphogenetic signaling pathways. Tapping physiological relevance, we show that m6A loss profoundly alters these cues and perturbs cellular fate choices and tissue architecture in all skin lineages. By single-cell transcriptomics and bioinformatics, both signaling and canonical translation pathways show significant downregulation after m6A loss. Interestingly, however, many highly m6A-modified mRNAs are markedly upregulated upon m6A loss, and they encode RNA-methylation, RNA-processing and RNA-metabolism factors. Together, our findings suggest that m6A functions to enhance translation of key morphogenetic regulators, while also destabilizing sentinel mRNAs that are primed to activate rescue pathways when m6A levels drop.

**\*For correspondence:**
fuchslb@rockefeller.edu

[†]These authors contributed equally to this work

## Introduction

Sophisticated gene expression regulatory machineries are necessary to achieve proper fate choices during mammalian embryogenesis. While transcriptional regulation has been studied in great depth, much less is known about how post-transcriptional regulation comes into play in tissue formation. Being an important element of post-transcriptional regulation of gene expression, chemical modifications on RNAs have been demonstrated to affect a wide range of RNA bio-activities. Among the over 100 types of RNA modifications identified thus far, m6A is the most abundant modification on mRNAs. Previous studies have identified m6A methyltransferases (writers), m6A demethylases (erasers) and also factors that recognize m6A-modified RNAs (readers) (*Alarcón et al., 2015*; *Bokar et al., 1997*; *Liu et al., 2014*; *Meyer et al., 2015*; *Meyer and Jaffrey, 2017*; *Patil et al., 2016*; *Zheng et al., 2013*; *Figure 1A*).

A number of studies have focused on dissecting m6A's functions in cell culture systems, where transcriptome-wide m6A profiles and expression patterns of the writers, erasers and readers were found to vary among cell types (*An et al., 2020*; *Delaunay and Frye, 2019*; *Roundtree et al., 2017*). In recent years, models have been made to conditionally disrupt specific m6A writers, erasers or readers in mouse tissues (*Cheng et al., 2019*; *Geula et al., 2015*; *Hsu et al., 2017*; *Ivanova et al., 2017*; *Lee et al., 2019*; *Li et al., 2017*; *Li et al., 2018*; *Lin et al., 2017*; *Shi et al., 2018*; *Wu et al., 2018*; *Xu et al., 2017*; *Yoon et al., 2017*; *Zhang et al., 2018*). While uniformly underscoring the physiological importance of m6A, these studies have also unveiled a profound

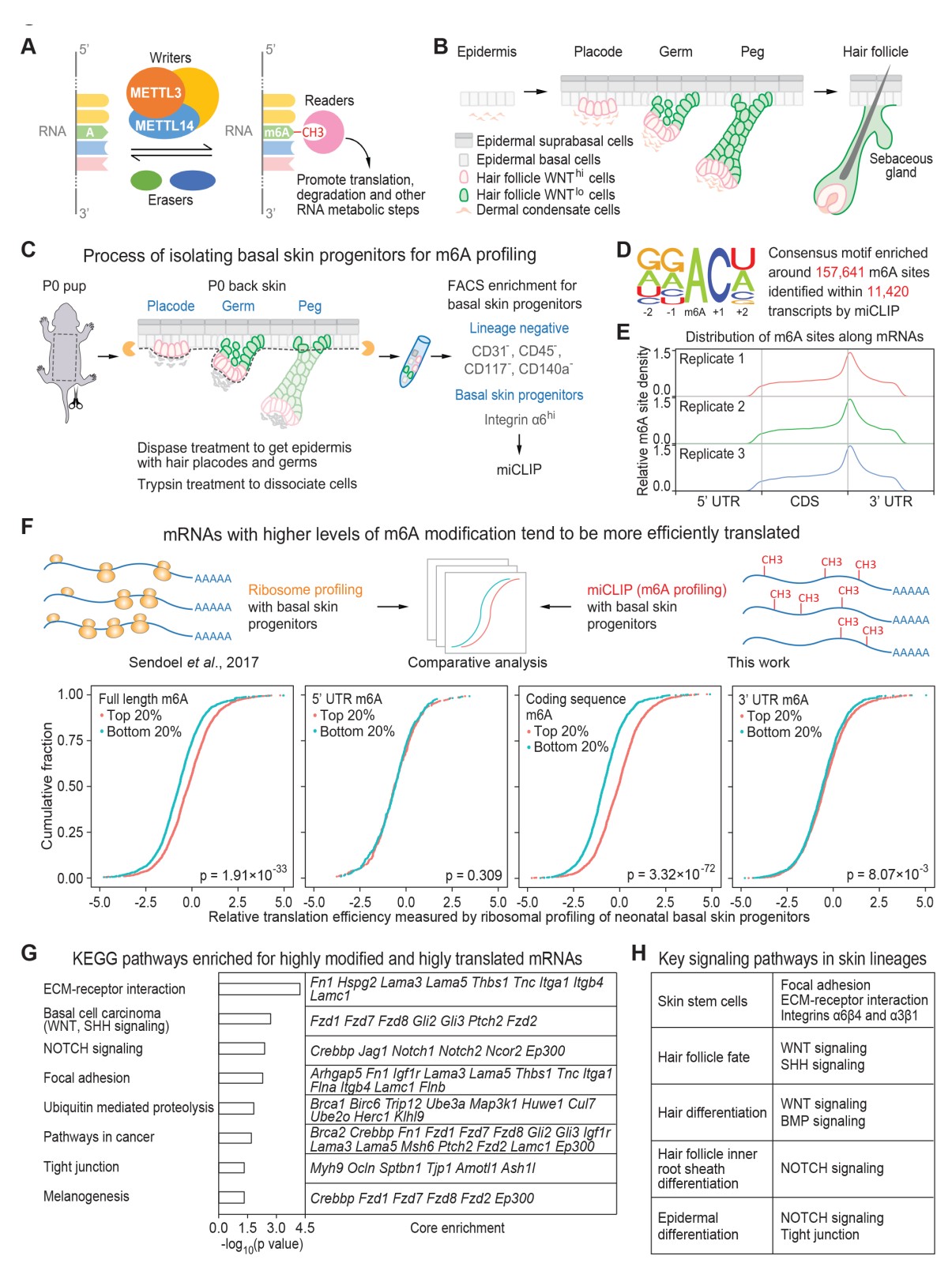

**Figure 1.** miCLIP and ribosomal profiling analyses of the mouse skin epithelial progenitors. (**A**) Schematic depicting the major factors involved in regulating the cellular dynamics of m6A modification. (**B**) Schematic depicting embryonic development of mammalian epithelial skin progenitors. WNT^hi/lo implies cells that show strong WNT or low WNT signaling as judged by *Axin2-LacZ* transgene expression (**Matos et al., 2020**). HF morphogenesis occurs in temporal waves, with mature HFs emerging shortly after birth. (**C**) Schematic depicting the process for enzymatically isolating

*Figure 1 continued on next page*

*Figure 1 continued*

epithelial progenitors from mouse skin at age P0. For miCLIP, cells were subjected to FACS purification as described in the Materials and methods. (**D**) Consensus sequence motif enriched around miCLIP-identified m6A sites in P0 skin progenitor mRNAs. (**E**) Metagene plots depicting the distribution of miCLIP-identified m6A sites along mRNAs. Data from three independent replicates are shown. (**F**) Schematic depicts comparison of the miCLIP data, which measures m6A modification to the ribosome profiling data, which landscapes bound ribosomes on neonatal skin progenitor mRNAs. The empirical cumulative distribution function (ECDF) plots compare the relative mRNA translation efficiency of the top 20% and bottom 20% of m6A-modified mRNAs. The data reveal that transcripts with higher levels of m6A modification (assessed by the sum of normalized-to-input uTPM value of m6A along the full-length transcript) tend to have higher levels of translation efficiency. The correlation between translation efficiency and the sum of normalized-to-input uTPM value of m6A at different regions of the mRNAs (5' UTR, coding sequence, 3' UTR) shows that the coding sequence m6A gives the best correlation to translation efficiency. The p values were calculated through Wilcoxon rank sum test. (**G**) GSEA of the overlap between mRNAs that are the top 20% heavily m6A-modified in coding sequence (assessed by the sum of normalized-to-input uTPM value of m6A along coding sequence) and the top 20% most efficiently translated mRNAs (assessed by ribosome profiling). Shown are the top eight enriched KEGG signaling pathways, each of which has a p value <0.05 and >5 enriched mRNAs. (**H**) Pathways known to play essential roles in regulating skin lineage specification.

The online version of this article includes the following figure supplement(s) for figure 1:

**Figure supplement 1.** miCLIP experiment setup and correlation analysis.

complexity in m6A's biological functions (*Du et al., 2016*; *Kennedy et al., 2016*; *Meyer et al., 2015*; *Wang et al., 2014a*; *Wang et al., 2015*; *Zaccara and Jaffrey, 2020*). Thus in some cell types and contexts, m6A modification promotes differentiation (*Geula et al., 2015*; *Lee et al., 2019*), while in others, it blocks differentiation (*Li et al., 2017*; *Vu et al., 2017*). Sometimes, as in the case of *Myc*, such opposing effects can be observed in different cell types sharing a common m6A target (*Lee et al., 2019*; *Vu et al., 2017*; *Yao et al., 2018*; *Zou et al., 2019*). In other cases, m6A targets can differ based upon cellular contexts and expression. Finally, m6A can exert context-dependent effects on distinct steps of RNA metabolism, compounding the unpredictability of whether a modification will impart to a given target increased or decreased expression.

The skin epithelium is an excellent model to begin to unravel the mechanisms underlying m6A's influences on various morphogenetic processes. Embryonic skin epithelium begins as a single layer of multipotent epithelial progenitors, which will develop into three strikingly distinct tissues: epidermis, hair follicles (HFs) and sebaceous glands (*Figure 1B*). Launching tissue diversification is WNT signaling, which begins heterogeneously within the progenitor population.

WNTs play a critical role in specifying the HF fate (*Andl et al., 2002*; *Gat et al., 1998*; *van Genderen et al., 1994*; *Huelsken et al., 2001*; *Xu et al., 2015*). Within the basal epidermal plane, WNT^hi embryonic progenitors cluster into hair placodes (*Ahtiainen et al., 2016*), which emerge in three waves beginning at embryonic day 14.5 (E14.5) and develop into fully mature HFs by birth (*Duverger and Morasso, 2009*; *Figure 1B*). The first divisions in WNT^hi placode cells are asymmetric, generating WNT^lo daughters that will give rise to hair follicle stem cells (HFSCs) and outer root sheath, and WNT^hi daughters that will generate the inner root sheath and hair shaft of the follicles (*Ouspenskaia et al., 2016*). Near birth, oil-rich sebaceous glands spawn from the upper portion of each HF (*Figure 1B*). Sebaceous gland development is favored over HFs in mice whose skin expresses ΔNLEF1, a dominant-negative disrupter of WNT signaling (*Merrill et al., 2001*; *Niemann et al., 2003*), indicating that WNT levels play an important role in balancing these fate choices.

Lower levels of WNT signaling also favor epidermal over HF fates, as in the developing plane of embryonic progenitors, those with lower levels of WNT signaling become fated to become epidermal progenitors, which fuel the production of upward columns of terminally differentiated cells that form the skin's barrier that excludes pathogens and retains body fluids. This differentiation program is regulated by NOTCH signaling and MYC activation (*Blanpain et al., 2006*; *Frye et al., 2003*; *Gandarillas and Watt, 1997*; *Moriyama et al., 2008*; *Rangarajan et al., 2001*; *Watt et al., 2008a*; *Watt et al., 2008b*).

While the levels of external signals and their downstream transcriptional effectors function critically in these epithelial fate choices within the skin, post-transcriptional regulation, such as translational control, has received more emphasis on balancing proper homeostasis once the tissue fate has been selected. Similar to the hematopoietic system (*Buszczak et al., 2014*; *Signer et al., 2014*), skin progenitors maintain reduced levels of protein synthesis relative to their differentiating progeny

(*Blanco et al., 2016*; *Liakath-Ali et al., 2018*; *Sendoel et al., 2017*). In differentiating skin cells, 5-methylcytosine (m5C) is enhanced, which protects tRNAs from cleavage and facilitates protein translation (*Blanco et al., 2011*). Under conditions of stress in the epidermis, EIF2α is phosphorylated, dampening protein translation, while a less efficient, more promiscuous initiator, EIF2A, translates key proteins that allow progenitor survival in a harsher environment (*Sendoel et al., 2017*). Interestingly, however, when translational control is compromised in the skin, epidermal, HF and sebaceous gland stem cell compartments respond differently, with epidermal cells implementing a feedback mechanism to increase global translation (*Liakath-Ali et al., 2018*). While context again surfaces as being key in triggering these varied responses, their purpose appears to be rooted in restoring homeostasis to the tissue.

How signaling, transcriptional regulation and cellular fate choices are linked to translational control and tissue homeostasis has remained elusive. Here, we show that m6A modifications of mRNAs may be involved in this connection. We began by mapping m6A at single-nucleotide resolution in skin epithelia. Exploiting our prior in vivo epidermal ribosomal profiling data (*Sendoel et al., 2017*), we then interrogated how m6A correlates with overall translational efficiency. Probing the physiological relevance of our findings, we employed genetics to conditionally ablate the deposition of m6A on RNAs. After analyzing the striking and differential consequences of m6A loss to the fates of the skin epithelial progenitors in vivo, we turned to single-cell RNA sequencing and functional studies to gain insights into the major cellular activities that are targeted by this regulatory machinery, and the significance of m6A to these three diverse programs of epithelial morphogenesis in the skin.

## Results

### Skin transcripts most highly modified by m6A are involved in HF morphogenesis

To assess the location and extent of m6A modification on the mRNAs expressed in mouse epidermal cells under physiological conditions, we performed m6A individual-nucleotide-resolution cross-linking and immunoprecipitation (miCLIP) (*Grozhik et al., 2017*; *Linder et al., 2015*) on poly(A)+ RNAs directly isolated from basal skin progenitors in vivo (*Figure 1C*). Adapting a procedure previously used for in vivo ribosomal profiling of these cells (*Sendoel et al., 2017*), we applied dispase treatment on newborn (postnatal day 0, P0) skin, which effectively removes the dermis, elongated hair pegs and HFs (*Rhee et al., 2006*; *Figure 1—figure supplement 1A*). This enabled us to apply fluorescence-activated cell sorting (FACS) for integrin α6 (CD49f) to enrich for progenitors of epidermis and hair placodes/germs (*Figure 1C*; *Figure 1—figure supplement 1B*). Sequencing libraries were then constructed from the m6A-immunoprecipitated mRNAs as well as the corresponding input RNA samples (*Figure 1—figure supplement 1C*).

Based upon the crosslinking-induced mutations detected in the miCLIP libraries, 11,420 transcripts displayed a total of 157,641 sites that were modified by m6A and typified by a 'DRACH' motif (*Fu et al., 2014*; *Linder et al., 2015*; *Figure 1D*; *Supplementary file 1*). The data were highly reproducible across three independent replicates, and as previously noted, m6A density was highest around mRNA stop codons (*Figure 1E*). To quantify the extent of m6A modification at each site, we calculated the unique tag counts per million (uTPM) as described in *Patil et al., 2016* around each site in the miCLIP data. We then normalized to the uTPM around the same site in the corresponding input data. This normalized-to-input uTPM value was also highly consistent across biological replicates (*Figure 1—figure supplement 1D*).

Exploiting our prior in vivo ribosome profiling data of mouse skin epidermal progenitors (*Sendoel et al., 2017*), we next examined in an unbiased fashion whether the degree of m6A modification of an mRNA correlated with its translational efficiency (*Figure 1F*, schematic). For every m6A-modified mRNA, we calculated the sum of normalized-to-input uTPM at each m6A site identified on full-length, 5' UTR, coding sequence, and 3' UTR. For each, we then compared the translational efficiencies of the top 20% most highly and bottom 20% least m6A-modified mRNAs from each way of calculation. Intriguingly, mRNAs with high levels of m6A modification tended to be more highly translated than those with low modification (*Figure 1F*). This correlation was most striking when we focused on m6A modifications in the coding sequence.

We then ranked the m6A-modified mRNAs according to the sum of normalized-to-input uTPM at each m6A site within the coding sequence (*Supplementary file 2*). Gene set enrichment analyses (GSEA) upon that value featured 'basal cell carcinoma', dominated by WNT and SHH signaling factors, and 'NOTCH signaling' among the top three categories (*Figure 1—figure supplement 1E,F*, *Supplementary file 2*). Interestingly, these top three categories tended to be more efficiently translated than other pathways (*Figure 1—figure supplement 1G*). Overall, these findings pointed to a potential importance of m6A modifications specifically within the coding sequence in promoting translation of an mRNA.

We then performed GSEA to interrogate the KEGG pathways enriched with the top 20% most highly modified and highly translated mRNAs. We focused on pathways with p values <0.05 and >5 enriched mRNAs/category. Although we do not discount the potential importance of any individual mRNA, it was easier to predict this potential for mRNAs with high coding sequence m6A levels, high translation and residing within larger enrichment categories.

The top categories sharing these criteria were signaling pathways including ECM-receptor interactions, basal cell carcinoma (dominated by mRNAs encoding WNT and SHH signaling factors) and NOTCH signaling (*Figure 1G*; *Supplementary file 2*). These pathways play essential roles in HF morphogenesis (*Figure 1H*). Notably, WNT signaling, which prompts basal progenitors to organize into hair placodes, precedes SHH signaling (*Woo et al., 2012*), and WNT signaling is required to maintain SHH pathway-driven basal cell carcinomas in mice and in humans (*Sánchez-Danés et al., 2018*).

Given the large number of mRNAs that were modified by m6A, we did not expect nor did we see a single category with an extraordinarily high enrichment score. However, the fact that the WNT and SHH pathways surfaced in multiple top enrichment categories was promising and suggested that m6A might function in promoting the HF fate.

## Conditional ablation of *Mettl3* in epidermal progenitors results in a marked defect in HF morphogenesis

The METTL3-METTL14 writer complex is responsible for adding the vast majority of m6A onto RNAs (*Figure 1A*). Upon ablation of *Mettl3* or *Mettl14,* global m6A is dramatically diminished in a wide variety of cell types (*Batista et al., 2014*; *Cheng et al., 2019*; *Cui et al., 2017*; *Geula et al., 2015*; *Lin et al., 2017*; *Liu et al., 2014*; *Vu et al., 2017*; *Wang et al., 2014b*; *Weng et al., 2018*; *Yao et al., 2018*; *Yoon et al., 2017*). Thus, to investigate the impact of m6A deficiency on skin morphogenesis, we generated *Mettl3* conditional knockout (cKO) mice harboring the following alleles: *Mettl3*$^{fl/fl}$, *Rosa26-YFP*$^{fl/+}$, *Krt14-Cre* (*Figure 2—figure supplement 1*). Active exclusively in the epidermal progenitors beginning at E13.5 (*Vasioukhin et al., 1999*), *Krt14-Cre* had excised the stop codon in the *Rosa26* locus by E14.5 (*Figure 2A*). Whole-mount immunofluorescence imaging revealed that by E16.5, expression of METTL3 was largely abolished within the targeted YFP$^+$ cells in the cKO skin epithelium (*Figure 2B*). And consistent with its known location in other cell types (*Kwon et al., 2019*; *Liu et al., 2014*; *Schöller et al., 2018*; *Zhang et al., 2018*), METTL3 was predominantly nuclear in control (Ctrl) epithelial progenitors.

We confirmed the quantitative loss of m6A by thin-layer chromatography (TLC), which we performed on poly(A)+ RNA isolated from the FACS-purified YFP$^+$ epidermal cells of *Mettl3* wild-type, heterozygous and null skin epithelium (*Figure 2C*). Quantification of the m6A/A ratio based on the radioactive signals revealed a comparable m6A content on mRNAs from wild-type *Mettl3*$^{+/+}$ and heterozygous *Mettl3*$^{+/-}$ cells, indicating that one allele of *Mettl3* was enough to sustain a normal m6A level on mRNAs. Thus from this point forward, we used heterozygous *Mettl3* $^{fl/+}$, *Rosa26-YFP*$^{fl/+}$, *Krt14-Cre* mice as our control.

Throughout embryogenesis, control and cKO mice were grossly comparable in appearance (*Figure 2D*). From postnatal day 2 (P2) onward, however, phenotypic abnormalities of the cKO animals began to emerge. Most striking was the progressive decline in body sizes and weights compared to their control littermates (*Figure 2—figure supplement 2A,B*). Most cKO animals did not survive beyond postnatal day 6 (P6).

To investigate potential causes of the lethality, we assessed the epidermal barrier function by measuring the trans-epidermal water loss (TEWL) rate of the control and cKO skin at P0 and P6. No significant TEWL differences were observed, ruling against dehydration as a cause of death (*Figure 2—figure supplement 2C*). Examination of oral tissues indicated that both teeth and tongue formed, but tongue filiform papillae, necessary for suckling and food intake at this stage, were

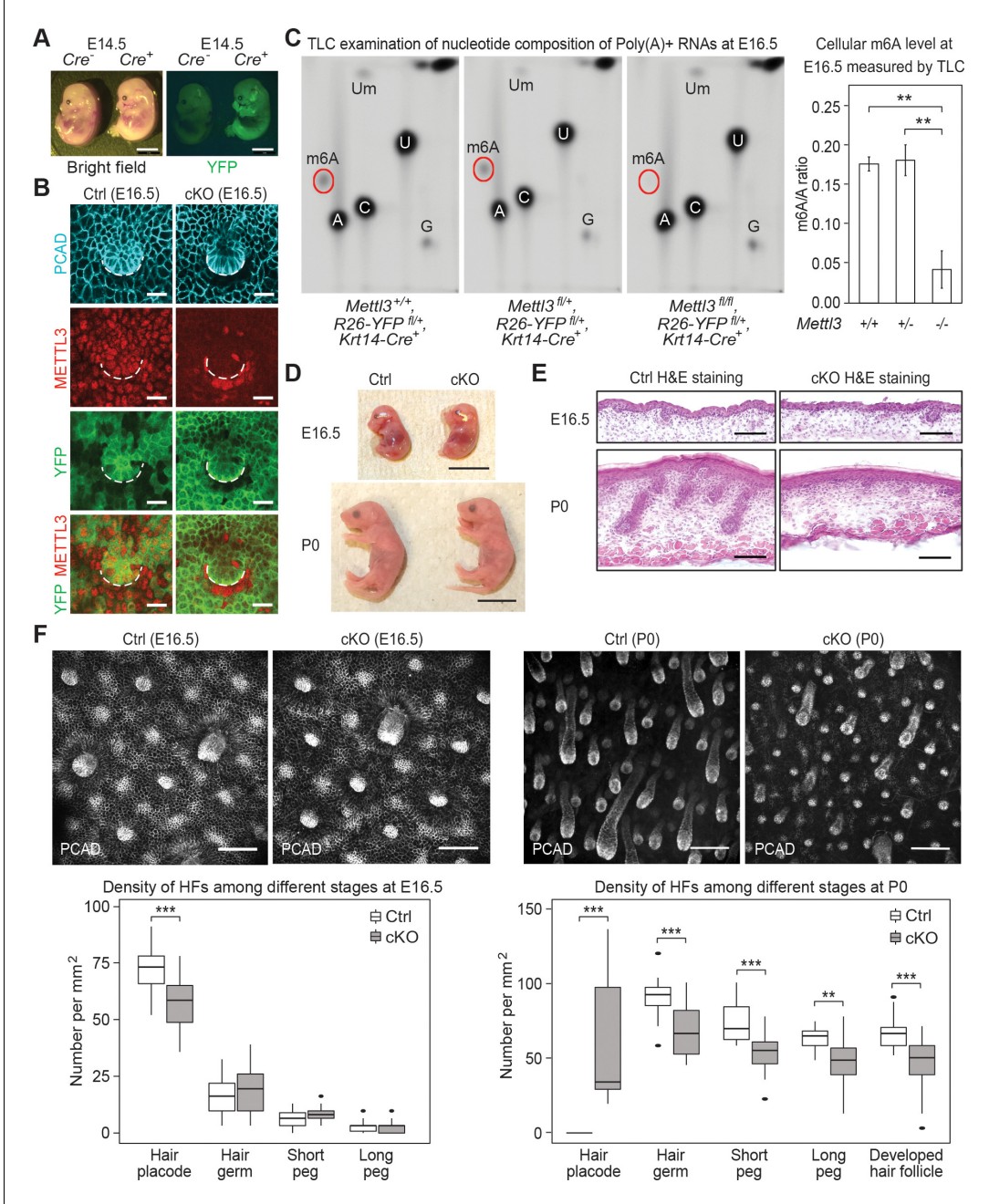

**Figure 2.** *Krt14-Cre* driven conditional *Mettl3* knockout mice display severe defects in HF morphogenesis. (**A**) Representative pictures of *Krt14-Cre*[-/-], *Rosa26-YFP*[fl/+] and *Krt14-Cre*[+/-], *Rosa26-YFP*[fl/+] littermate embryos demonstrating the onset of uniform YFP expression in the E14.5 skin epithelium (scale bars: 2 mm). (**B**) Confocal images of E16.5 whole-mount back skin immunolabeled for P-cadherin (PCAD), METTL3 and YFP (scale bars: 20 μm). Note that nuclear METTL3 immunofluorescence is selectively depleted from the YFP[+] cells in cKO skin. White dashed lines denote the dermal-epidermal border. (**C**) Left panel: representative pictures of thin layer chromatography (TLC) on Poly(A)+ RNA samples isolated from E16.5 skin epithelial cells. Right panel: quantification of m6A levels based on TLC (error bars: standard deviation, for *Mettl3*[+/+] n = 2 biological replicates, for each of the other conditions n = 3 biological replicates, **p<0.01 by unpaired two-tailed Student's t-test). (**D**) Representative pictures of E16.5, P0 control and cKO littermates (scale bars: 1 cm). (**E**) Hematoxylin and eosin (H and E) stained back skin sagittal sections at indicated time points (scale bars: 100 μm). (**F**) Confocal images of whole-mount back skin immunolabeled for PCAD at indicated time points (scale bars: 100 μm) and quantification of HFs at different developmental stages (for E16.5, n = 3 biological replicates ×10 images per replicate and for P0, n = 18 images from four biological replicates, **p<0.01 and ***p<0.001 by unpaired two-tailed Student's t-test).

The online version of this article includes the following source data and figure supplement(s) for figure 2:

**Source data 1.** Quantification of m6A/A ratio through TLC signals in (C).

*Figure 2 continued on next page*

eLife Research article

Developmental Biology

*Figure 2 continued*

**Source data 2.** Quantification of HF density in (F).
**Figure supplement 1.** Breeding strategy to generate the *Mettl3* cKO animals and the control (Ctrl) littermates.
**Figure supplement 2.** Additional information on the *Mettl3* cKO phenotypes.
**Figure supplement 2—source data 1.** Quantification of neonates' body weights in (B).
**Figure supplement 2—source data 2.** Quantification of TEWL in (C).

notably diminished (*Figure 2—figure supplement 2D,E*). Back skin HF morphogenesis was also severely altered in the cKO animals. This began to surface soon after *Mettl3* ablation where the numbers of de novo placodes began to decline, and progressed thereafter as judged by the paucity of hair germs and pegs at birth (*Figure 2E,F*). Analyses of the few pups still alive by P6 revealed that with the exception of the sparse, large guard hairs, which form prior to *Krt14-Cre* activity and do not use WNT signaling in their specification, HF downgrowth was largely impaired and HFs failed to mature beyond the peg stage (*Figure 2—figure supplement 2F*).

Based upon these data, the defects did not appear to reflect developmental delays, but rather altered morphogenesis. In this regard, the severity of defects in the tongue filiform papillae and skin HFs was particularly intriguing as both require WNT and SHH signaling for their morphogenesis (*DasGupta and Fuchs, 1999*; *Iwatsuki et al., 2007*; *Järvinen et al., 2006*). At the top of these morphogenetic cascades is LEF1, the transcription factor which binds to WNT effector β-catenin and translocates to the nucleus, where it is required to mediate WNT signaling and launch cellular fates (*Adam et al., 2018*; *van Genderen et al., 1994*; *Ouspenskaia et al., 2016*; *Zhou et al., 1995*).

Immunofluorescence revealed that at E17.5, nuclear LEF1 was significantly diminished in the basal (placode) progenitors within the epidermal plane as well as WNT$^{hi}$ cells in the developing HFs of cKO skin (*Figure 3A*). At P0, LEF1 levels remained low in the WNT$^{hi}$ cells. It was also diminished in the underlying dermal condensates (precursors to the dermal papilla, DP), whose WNT signaling and nuclear LEF1 is known to be dependent upon epithelial WNT signaling (*Mok et al., 2019*; *Figure 3B*). Notably, immunohistochemistry revealed that in contrast to control HFs, which showed appreciable anti-β-catenin nuclear staining at the follicle:DP interface, in *Mettl3* null HFs, β-catenin was mostly at the intercellular borders, reflective of its WNT-independent role at adherens junctions (*Figure 3C*). Additionally, these HF progenitors were perturbed in engulfing the DP, a feature seen when WNT signaling and/or SHH signaling are inhibited (*Heitman et al., 2020*; *Matos et al., 2020*).

Since SHH signaling is dependent upon WNT signaling, we next tested for the activation of this pathway. To do so, we mated our mice to *Gli1-lacZ* mice, and performed X-gal histochemical staining on skin sections. While SHH signaling was not blocked in the absence of METTL3, it was aberrant, with an overall reduction, particularly in the epithelium (*Figure 3D*). These results were intriguing in light of the knowledge that laminin 5-1-1 is required for HF downgrowth and SHH signaling (*Fleger-Weckmann et al., 2016*; *Gao et al., 2008*), and that *Lama5* and *Lamac1* were among the most highly modified and efficiently translated mRNAs in skin (*Figure 1G*).

As an appendage of HFs, sebaceous glands use a different hedgehog pathway and require reduced WNT signaling for their formation (*Merrill et al., 2001*; *Niemann et al., 2003*). Consistent with diminished WNT signaling in the *Mettl3* cKO skin, an increase in Oil Red O staining was seen in the aberrant HFs (*Figure 3E*). The paucity of lipids in the P6 dermis was expected due to the failure to suckle and weight loss likely caused by the tongue defects. These phenotypes bore a striking resemblance to the mice expressing a dominant negative form of LEF1 in their skin epithelium.

To interrogate the long-term consequences of epidermal m6A depletion, we engrafted the back skin of P0 control and cKO littermates onto *Nude* mice. By the 15th day post-engraftment, the epithelium of both control and cKO skins were YFP$^+$, indicating that they had survived the engraftment procedure (*Figure 3—figure supplement 1*). However, whereas control skin had generated a thick hair coat, the cKO skin showed a striking paucity of hair growth (*Figure 3F,G*). Oil Red O staining further corroborated our P6 analyses and suggested that cKO progenitors follow the sebocyte versus HF path of morphogenesis (*Figure 3H*).

Many of these features were confirmed at the ultrastructural level, including the arrested growth of HFs and the presence of *Mettl3* null HFs that failed to engulf the DP (*Figure 4A*). Although we did not see gross signs of junctional defects, we did see morphological signs of apoptosis in P0 HFs,

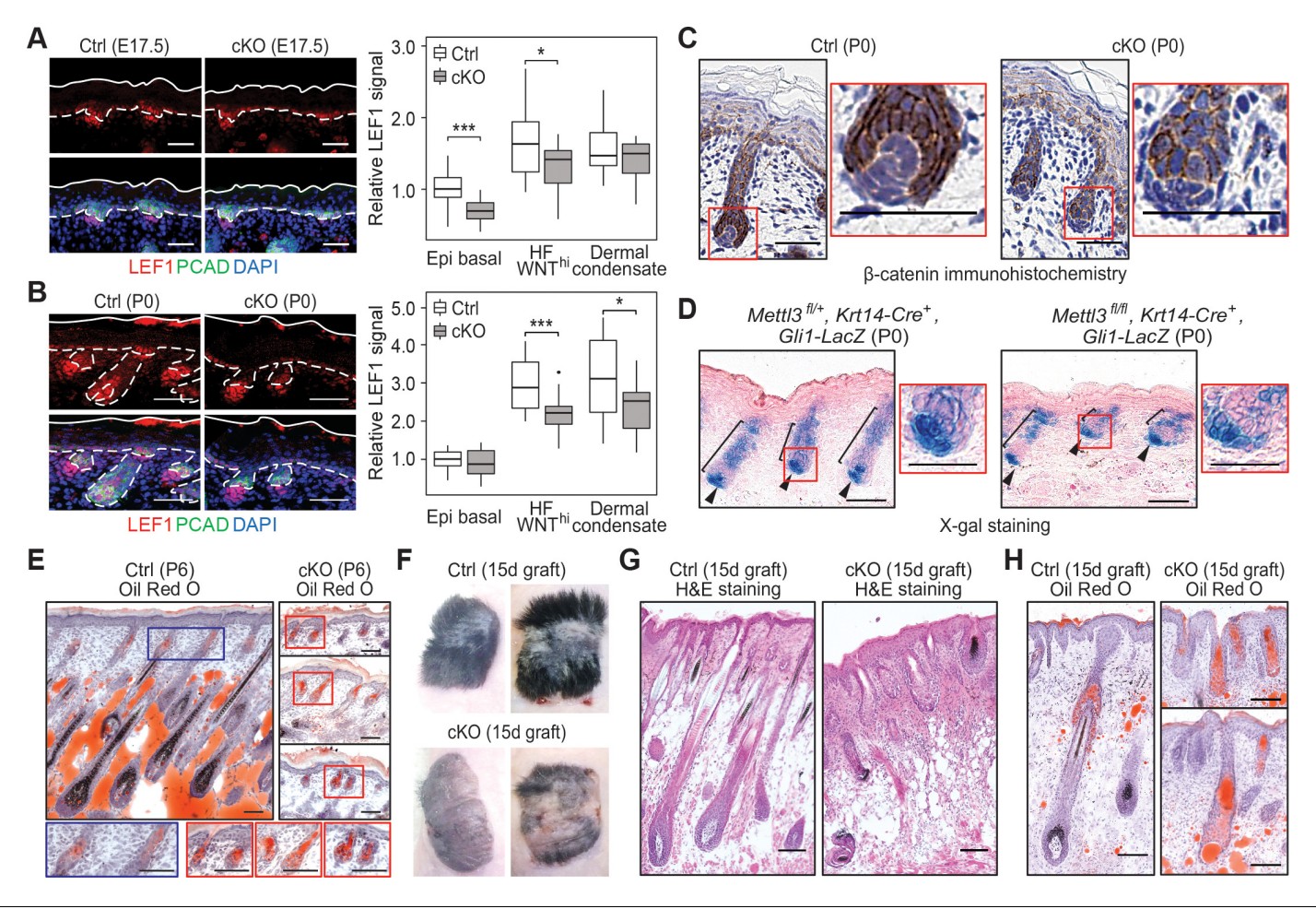

**Figure 3.** Loss of m6A results in diminished WNT signaling and signs of perturbed HF fate. (**A**) Left panel: representative images from E17.5 sagittal sections immunolabeled for LEF1 and PCAD (scale bars: 50 μm). White solid lines denote skin surface and dashed lines denote dermal-epidermal border. Right panel: quantification of LEF1 immunofluorescence at indicated compartments (for each condition n = 3 biological replicates ×7 images per replicate, *p<0.05 and ***p<0.001 by unpaired two-tailed Student's t-test). (**B**) Left panel: representative images from P0 sagittal sections immunolabeled for LEF1 and PCAD (scale bars: 50 μm). Solid and dashed lines as in (**A**). Right panel: quantifications of LEF1 immunofluorescence in indicated compartments (for each condition n = 3 biological replicates ×7 images per replicate, *p<0.05 and ***p<0.001 by unpaired two-tailed Student's t-test). (**C**) Representative images from P0 sagittal sections with immunohistochemistry staining of β-catenin, counter-stained with hematoxylin (scale bars: 50 μm). (**D**) Representative images from P0 sagittal sections stained with X-gal and nuclear fast red to examine the expression of the *Gli1-LacZ* transgene, a proxy for SHH signaling (scale bars: 100 μm). (**E**) Representative images from P6 sagittal sections stained with Oil Red O and counterstained with hematoxylin to visualize signs of HF to sebocyte fate switching within the epithelium (scale bars: 100 μm). The staining in the control dermis reflects adipocyte-derived lipids, missing in the cKO pups, which lose weight after birth. (**F**) Representative pictures of control (Ctrl) and cKO back skins engrafted onto *Nude (Nu/Nu)* mice and analyzed 15 days later. (**G**) Representative Hematoxylin and eosin (H and E) stained sagittal sections of 15-dayengrafted back skins (scale bars: 100 μm). (**H**) Representative sagittal sections as in (**G**) and counterstained with Oil Red O and hematoxylin (scale bars: 100 μm).

The online version of this article includes the following source data and figure supplement(s) for figure 3:

**Source data 1.** Quantification of LEF1 immunofluorescence signals at E17.5 in (A).
**Source data 2.** Quantification of LEF1 immunofluorescence signals at P0 in (B).
**Figure supplement 1.** Examination of METTL3 and YFP expression in the grafted skin.

which we substantiated by activated (cleaved) Caspase-3 and TUNEL staining (*Figure 4—figure supplement 1A*), suggesting that the defects went beyond mere signaling perturbations. We also noted *Mettl3* null HF cells that were positive for TUNEL, but not activated Caspase-3. While this is a feature of sebocytes (*Fischer et al., 2017*), it could also be a sign of general cell degradation. Moreover, the organization of HF cells also appeared to be perturbed. In the hair placodes that did initiate,

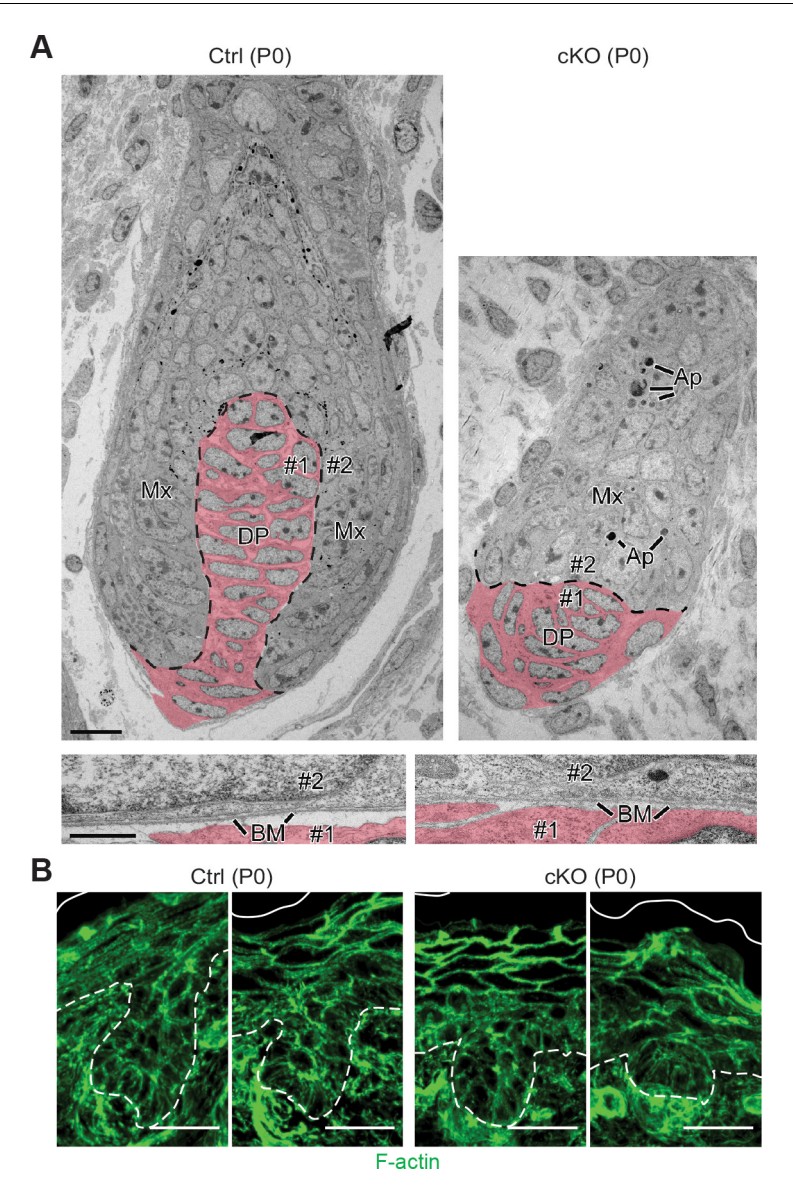

**Figure 4.** Perturbations in WNT-driven dermal papilla engulfment and in actin-mediated cellular polarity within *Mettl3* cKO HFs. (**A**) Ultrastructure of HF in control and *Mettl3* cKO mice in P0 back skin. Matrix (Mx) cells engulf the dermal papilla (DP, colored in pink) in the control HF but often fail to do so in the cKO. Dashed line indicates the boundary between matrix and dermal papilla. The boundary between dermal papilla cell #1 and matrix cell #2 is magnified in the panel below. BM, basement membrane. Scale bars: 10 μm (upper panel), 600 nm (lower panel). Ap, apoptotic bodies. (**B**) Representative images from P0 sagittal sections labeled for F-actin by phalloidin (scale bars: 25 μm). White solid lines denote skin surface and dashed lines denote dermal-epidermal border. Note apical polarization of F-actin in control HFs, often missing in *Mettl3* cKO follicles.

The online version of this article includes the following source data and figure supplement(s) for figure 4:

**Figure supplement 1.** Analysis of DNA fragmentation, apoptosis and cell division angles in HF morphogenesis.

**Figure supplement 1—source data 1.** Quantification of cell death events in HFs in (A).

**Figure supplement 1—source data 2.** Quantification of cell division angles in (B).

spindle orientations appeared to be largely unaltered and remained perpendicular to the underlying basement membrane (*Figure 4—figure supplement 1B*). However, as judged by phalloidin staining to visualize F-actin, apical cytoskeletal polarization of progenitors facing the DP appeared to be perturbed (*Figure 4B*).

## Loss of m6A results in perturbations within the single-cell transcriptomics of the HF lineage

To further interrogate how loss of METTL3 affects the skin epithelium, we first addressed whether we might be able to obtain sufficient *Mettl3* null progenitors to perform in vivo ribosomal profiling and proteomics/immunoblot analyses. We first tried dispase treatment to enrich for in vivo skin epithelium, but in contrast to control skin, cKO hair placodes/germs were left behind in the dermal fraction as early as E17.5 (*Figure 5—figure supplement 1A,B*). When we turned to in vitro culture, we learned that despite the ability of cKO skin progenitors to survive in vivo until birth, *Mettl3* null skin epithelial progenitors failed to form colonies (*Figure 5—figure supplement 1C*). Although the underlying basis for the puzzling differential sensitivity to dispase and inability to survive in vitro are likely complex and beyond the scope of the present study, it compromised our ability to perform ribosomal profiling or proteomics/immunoblot analyses on *Mettl3* cKO progenitors.

Although these limitations precluded high throughput analyses to interrogate the direct consequences of m6A loss to translation and protein production, we were still able to ascertain the consequences to global gene expression through single-cell RNA sequencing (scRNA-seq), an analysis requiring only a few thousand cells. We focused on E17, a time when METTL3 and transcriptome-wide m6A were lost, but before marked differences in the skin epithelium or its classical cell identity markers had surfaced (*Figure 5—figure supplement 1D*). We FACS-purified YFP$^+$ cells from trypsinized skins of control and *Mettl3* cKO embryos, and then binned intact YFP$^+$ cells according to their exclusion of 4', 6-diamidino-2-phenylindole (DAPI) staining and their relative levels of integrins (α6, β1) (*Figure 5—figure supplement 1E*). We then processed these isolated cells through the 10X Genomics Next GEM system for scRNA-seq, from which we obtained high-quality sequencing data from 4443 control cells and 3455 *Mettl3* null cells.

Unbiased clustering of the cells based on their gene expression profiles indicated seven major clusters as revealed by t-distributed stochastic neighbor embedding (t-SNE) plots whose identities were ascertained based upon established markers (*Figure 5A*, *Figure 5—figure supplement 1F*). Three clusters were related to basal epidermal progenitors (Epi basal) and were typified by high *Krt5* and *Krt14* expression. We classified one cluster as the differentiating suprabasal epidermal keratinocytes (Epi suprabasal), based upon their high *Krt1* and *Krt10* expression. As expected, some of these *Krt1*$^+$*Krt10*$^+$ cells were still cycling, a feature of the asymmetric cell divisions that occur at this stage of embryonic development and which place daughter cells from proliferative parents into the suprabasal layers (*Williams et al., 2011*). The cluster with elevated expression of *Shh*, *Lef1* and *Ctnnb1* (encoding WNT-activated β-catenin) was identified as the WNT$^{hi}$ cells of specified HFs, while the two clusters with high expression of *Krt17* and *Sox9* and little or no *Lef1* were typical of WNT$^{lo}$ cells.

A heat map of the expression of these markers and their clustering into these seven groups is presented in *Figure 5B*. The molecular distinctions among the three basal epidermal progenitor populations and among the two WNT$^{lo}$ populations appeared to at least partially reside within cell cycle stage differences, as judged by expression of cell cycle mRNAs such as *Cdk1* and *Mki67* (*Figure 5B*, *Figure 5—figure supplement 1F*). However, since markers such as SCA1 (*Ly6a*), that molecularly distinguish adult progenitors of the interfollicular epidermis from those of the upper outer root sheath were not yet expressed at this time, other differences in these signatures could represent early signs of their specification. Regarding the WNT$^{lo}$ populations, the *Lhx2*-expressing cells showed features more characteristic of early HF stem cells (*Rhee et al., 2006*), while the *Krt79*-expressing cells were likely to be early sebaceous gland progenitors (*Veniaminova et al., 2019*).

To assess lineage differentiation trajectories, we performed pseudotime analyses on both control and cKO transcriptomes (*Figure 5C*). As expected, for control skin, most of the cells fell along two main branches: a hair germ branch composed largely of WNT$^{hi}$ (*Krt17*$^+$*Lhx2*$^+$*Sox9*$^{neg}$) and WNT$^{lo}$ (*Krt17*$^+$*Sox9*$^+$*Tgfbr2*$^+$) cells; and an interfollicular epidermal branch of *Krt5*$^{hi}$*Krt14*$^{hi}$ basal progenitors and their suprabasal *Krt1*$^+$*Krt10*$^+$ progeny. In agreement with the phenotypic abnormalities, the population of WNT$^{hi}$ cells along the HF branch was markedly diminished in the cKO, and some WNT$^{hi}$ cells were abnormally positioned along an extra lineage branch. These findings were consistent with the reduced density of hair placodes seen at this time (*Figure 2F*).

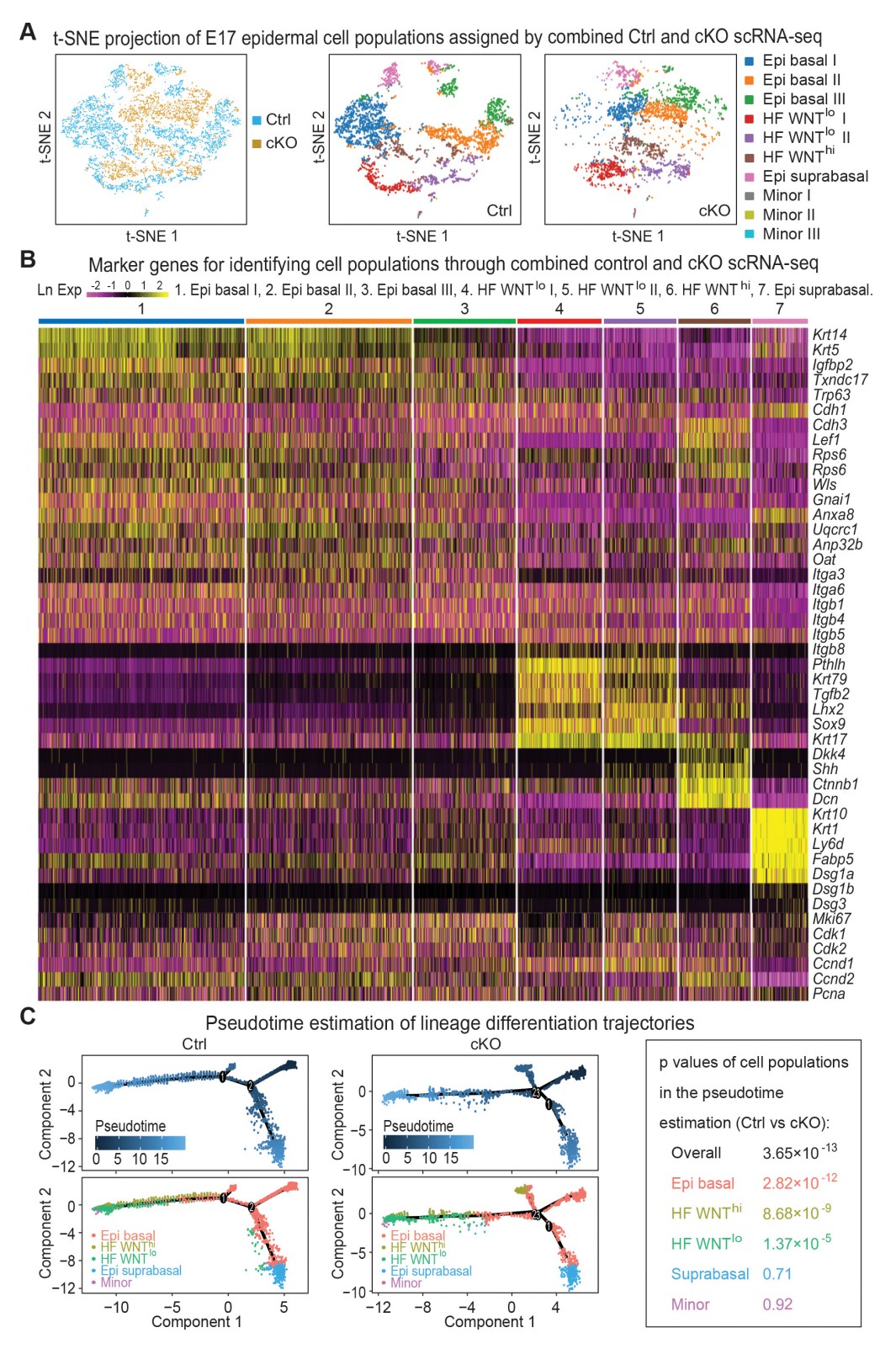

**Figure 5.** Single-cell transcriptomics of *Mettl3* cKO compared to control skin epithelial lineages. (**A**) YFP[+] progenitors were FACS-enriched from E17 control and *Mettl3* cKO whole back skins and subject to scRNA-seq as described in the Materials and methods. Shown are the data from unbiased clustering of the single cells projected by t-SNE. (Clustering was performed on all cells from control and cKO samples pooled together.) Seven major clusters and three minor clusters were identified, which are present in both samples. (**B**) Heat map illustrating how the seven populations were

*Figure 5 continued on next page*

*Figure 5 continued*

identified based upon their expression of established marker genes. Each column depicts data from one single cell belonging to the cluster that is indicated at the top. Each row illustrates cluster-wide expression data for the specific mRNA listed at the right. Color coding of each cluster is according to the color scheme in the t-SNE plots in (A). (C) Pseudotime plots showing the estimated lineage trajectories of each of the clusters as derived from the scRNA-seq data. The p values were calculated through Pearson's chi-square test.

The online version of this article includes the following figure supplement(s) for figure 5:

**Figure supplement 1.** Cell isolation for scRNA-seq analysis and cell identity verification.

## Signaling pathways and canonical translation initiation factors are among the mRNAs that are downregulated upon METTL3 loss

To gain further insights into m6A's function, we next evaluated how steady-state levels of each mRNA within a skin cell population changed upon METTL3 depletion (*Supplementary file 3*), and then correlated this change to m6A modification levels. In addition to the sum of normalized-to-input uTPM parameter that we used earlier, we also tested other parameters to assess m6A levels (*Supplementary file 4*). The degree of correlation with normalized-to-input uTPM was higher than merely counting m6A site numbers. Correlations were also higher when we normalized to RNA length (i.e. per nucleotide, per nt) and focused on the density of m6A within the coding sequence rather than the 5' or 3' UTRs or full-length mRNA (*Figure 6—figure supplement 1*). Therefore, in the following analysis with the scRNA-seq data, the per nt sum of normalized-to-input uTPM in coding sequence (coding sequence SN-uTPM per nt) parameter was used to evaluate the importance of m6A levels with regards to mRNA level changes arising from METTL3 loss.

Given our data thus far, we began by focusing on those RNAs that were downregulated upon METTL3 loss. Since HFs are specified within the basal epidermal plane and are WNT^hi, we primarily focused on the most significantly downregulated mRNAs in these categories. Although the dynamic range of fold changes for this RNA cohort was low, 2555 genes were downregulated with a cut-off Z score of $<-1.96$ (all dots on the left of the dotted line in the plots of *Figure 6A*; *Supplementary file 3*). Intriguingly, many of these mRNAs fell into signaling pathways that had surfaced when we discovered a correlation between high m6A with efficient translation (*Figure 6B*; compare with *Figure 1G*). Moreover, although the mRNAs within these categories differed, some of the most significantly downregulated mRNAs upon *Mettl3* ablation were ones that were amongst the most highly m6A-modified in wild-type cells (blue dots in *Figure 6A*; blue-highlighted mRNAs in *Figure 6B*). Of additional note, canonical translation initiation complex factors were also included in the cohort of significantly downregulated mRNAs, suggesting an additional layer of effects on global translation through m6A.

Given that a number of phenotypic features of METTL3 loss are recapitulated in WNT pathway mutants, we naturally gravitated toward the WNT signaling mRNAs that were significantly downregulated. Although *Lef1* was not within the top 20% of m6A-modified mRNAs, it was within the top cohort of significantly downregulated mRNAs upon *Mettl3* targeting and was confirmed by quantitative PCR (qPCR) (*Figure 6C*). *Lef1* mRNAs were reduced in both basal epidermal and WNT^hi progenitors, further suggesting that some basal epidermal progenitors were unable to progress to form HFs without m6A modification. Many other factors required for WNT-mediated cell fate specification were also significantly downregulated (*Figure 6B*). Although changes were relatively modest at this early stage, levels of WNT signaling are known to profoundly impact fate outcomes and proper tissue morphogenesis (*Buechling and Boutros, 2011*; *Tortelote et al., 2017*).

It was also notable that the NOTCH signaling pathway also resurfaced in our loss of function analyses of basal epidermal progenitors. Both *Jag1* and *Maml2* were among the most highly m6A-modified mRNAs and among the most significantly downregulated (*Figure 6B*). NOTCH signaling functions suprabasally at the transition between the basal and spinous layer, where it is typified by expression of classical NOTCH target, HES1 (*Blanpain et al., 2006*). Indeed, as judged by whole-mount immunofluorescence, HES1 was diminished (*Figure 6D*).

While HES1 downregulation correlated with the changes in gene expression revealed by scRNA-seq, it was also possible that it was reflective of a change in the flux of basal cells into the spinous layers. We were particularly intrigued by this latter possibility since components of hemidesmosomes, for example BPAG1 (*Col17A1*) and integrin β4 (*Itgb4*), which are responsible for the bulk of

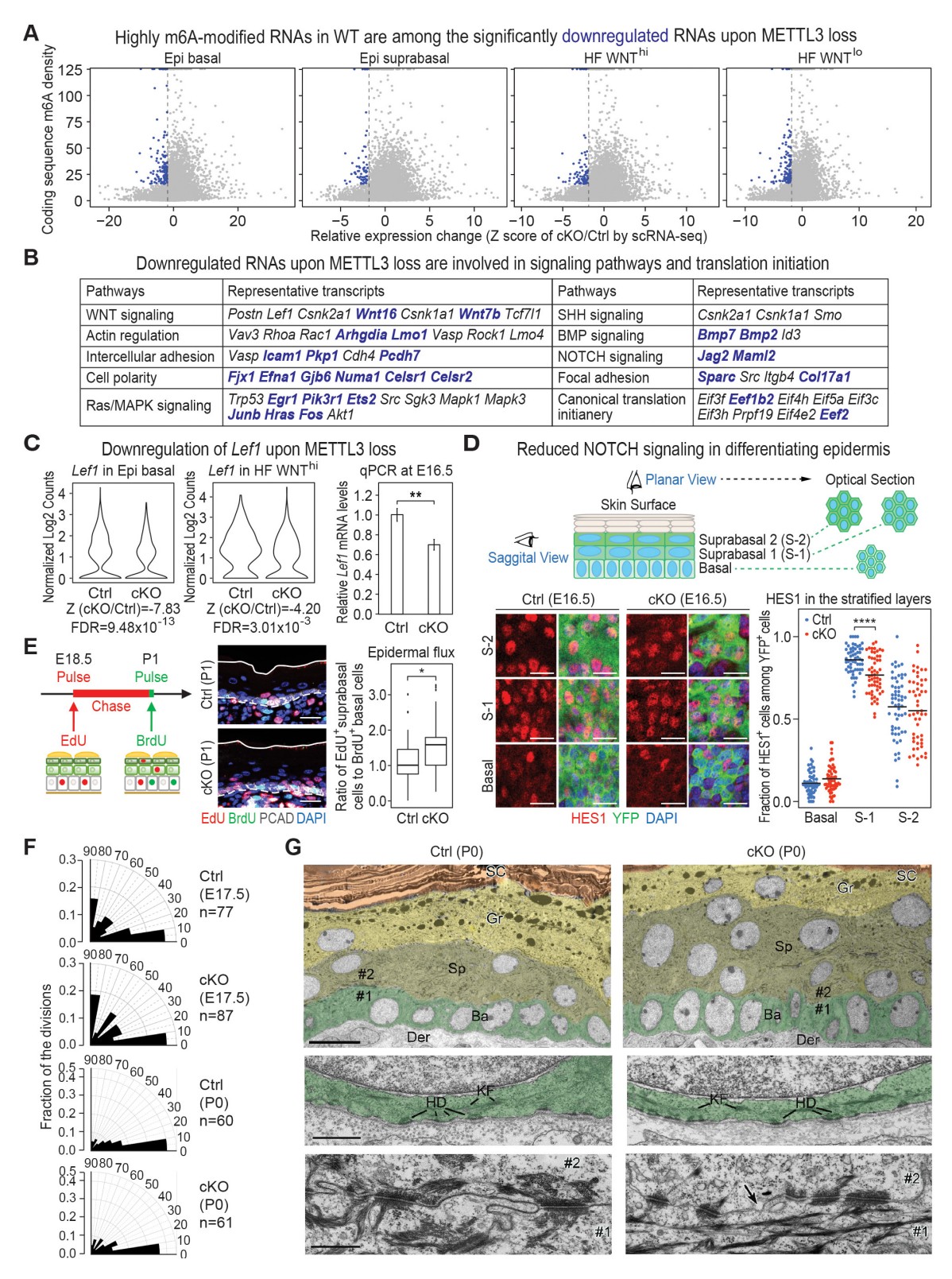

**Figure 6.** Investigation of RNAs whose levels diminish upon METTL3 loss. (**A**) scRNA-seq data from *Figure 5A* were binned according to four major classifications: Epi basal, Epi suprabasal, HF WNT^hi and HF WNT^lo. Scatter plots of mRNAs in these cells were then analyzed according to their expression changes (cKO/Ctrl) assessed by scRNA-seq Z score and to their coding sequence m6A density in wild-type (WT) skin epithelium assessed by the miCLIP SN-uTPM per nt value. Dots on the left of the dashed line in each plot indicate RNAs which from scRNA-seq have a Z score (cKO/Ctrl)

*Figure 6 continued on next page*

*Figure 6 continued*

<−1.96 (*Supplementary file 3*). Among those, blue dots denote mRNAs with m6A coding sequence SN-uTPM per nt among the top 20% (*Supplementary file 4*). (B) Major pathways and their associated RNAs that were downregulated upon METTL3 loss. Shown are the data for the basal epidermal progenitors, which at E17 contained both epidermal and hair placode cells. mRNAs highlighted in blue correspond to blue dots in (A), and were among the most significantly downregulated upon METTL3 loss but heavily m6A-modified in wild-type. Note that many of these pathways also corresponded to those whose heavily m6A-modified mRNAs were also efficiently translated. (C) Violin plots illustrating the relative expression levels of *Lef1* mRNA in the *Mettl3* cKO versus control basal epidermal progenitors and WNT$^{hi}$ progenitors. Z score assessment of expressional difference between cKO and control [Z (cKO/Ctrl)] and false discovery rate (FDR) is calculated by MAST. The down-regulation was verified with qPCR on total RNA samples extracted from YFP$^+$ skin epithelial cells FACS isolated from E16.5 embryos with *Tbp* mRNA as internal control (error bars: standard deviation, for each condition n = 3 biological replicates, **p<0.01 by unpaired two-tailed Student's t-test). (D) Confocal images of E16.5 whole-mount back skin immunolabeled for HES1 and YFP (scale bars: 20 µm). HES1 expression was quantified in the stratified layers of skin epithelium (middle line corresponds to the mean; for each condition, the data are from two biological replicates; ****p<0.0001 by unpaired two-tailed Student's t-test). (E) Pulse-chase assay examining the rate of epidermal cell flux from basal to suprabasal layers. Control and cKO animals were pulsed at E18.5 with EdU and the signal was then chased until P1. Before tissue collection, the P1 pups were treated with a short (1 hr) BrdU pulse. P1 back skin sagittal sections were subjected to immunofluorescence to examine the EdU and BrdU labeling in the basal versus suprabasal layers of the epidermis (scale bars: 25 µm). White solid lines denote skin surface and dashed lines denote dermal-epidermal border. Cell flux rates were quantified based on the ratio of EdU$^+$ cells in the suprabasal layer to BrdU$^+$ basal cells (for each condition n = 4 biological replicates ×10 images per replicate, *p<0.05 by unpaired two-tailed Student's t-test). (F) Radial histograms depicting the division orientation of epidermal basal cells during anaphase/telophase at E17.5 and P0, assessed by IF staining of Survivin, integrin β4 (CD104) and PCAD as described in *Williams et al., 2011*. For each condition, three biological replicates were analyzed and n indicates the total number of anaphase/telophase cells examined from the embryos. (G) Ultrastructure of epidermis in control and *Mettl3* cKO P0 back skin. Ba, basal layer, colored in green; Sp, spinous layer, colored in greenish yellow; Gr, granular layer, colored in yellow; SC, stratum corneum, colored in orange. Note the increased numbers of cells in the spinous layer of cKO, and the presence of nuclei in many cells of the granular layer. The boundary between dermis (Der) and the basal layer is shown in the middle panel. KF, keratin filaments; HD, hemidesmosomes. The border between cell #1 (basal) and cell #2 (suprabasal) is shown in the lower panel. Intercellular membranes are sealed in the control. Note small gaps (arrow) are present at the intercellular border, more frequently in cKO than in control. Scale bars: 10 µm (upper panel), 600 nm (middle and lower panel).

The online version of this article includes the following source data and figure supplement(s) for figure 6:

**Source data 1.** *Lef1* qPCR in (C).
**Source data 2.** Quantification of HES1 immunofluorescence signals in (D).
**Source data 3.** Quantification of EdU$^+$ and BrdU$^+$ cells in (E).
**Source data 4.** Quantification of cell division angles in (F).
**Figure supplement 1.** Correlation between the levels of m6A modification and changes in steady-state RNA levels upon *Mettl3* ablation.
**Figure supplement 2.** Additional analysis of epidermal perturbations upon *Mettl3* cKO.
**Figure supplement 2—source data 1.** Quantification of PCAD, ECAD immunofluorescence signals in (A).
**Figure supplement 2—source data 2.** Quantification of EdU$^+$ cells and the suprabasal/basal cell number ratio in (B).
**Figure supplement 2—source data 3.** Quantification of cell sizes by cytospin in (C).
**Figure supplement 2—source data 4.** Quantification of cell death events in epidermis in (F).

basal cell adhesion to the basement membrane (*Dowling et al., 1996*), were among the most significantly downregulated mRNAs upon METTL3 loss (*Figure 6B*). We therefore performed a pulse-chase experiment, by first administering a short pulse of deoxythymidine analog 5-ethynyl-2′-deoxyuridine (EdU) on E18.5 pups, and then treating with bromodeoxyuridine (BrdU) just prior to analyses (*Figure 6E*). Although labeling was somewhat higher in the basal cells of cKO compared to control skin, BrdU incorporation was limited to the basal progenitors in both. Most notably, the percentage of EdU-suprabasal:BrdU-basal cells was greater in the cKO skin, indicating that *Mettl3* null basal progenitors fluxed at a higher rate than normal into the suprabasal layers.

Additional signs of basal:suprabasal perturbations in the epidermis were revealed by immunofluorescence imaging for the basal progenitor cadherin, P-cadherin (PCAD), relative to E-cadherin (ECAD), a pan-epidermal marker. While fluorescence intensities showed that both cadherins were elevated at intercellular borders, the PCAD:ECAD ratio was significantly lower than normal (*Figure 6—figure supplement 2A*). PCAD is often associated with cells that undergo migration and invasion, while ECAD is typically associated with more static intercellular contacts (*Kümper and Ridley, 2010*). Thus, the reduction in PCAD:ECAD ratio, along with the diminished actin regulators, were further consistent with enhanced departure of *Mettl3* null cells from the basal layer, where cells have traction.

The spindle orientations of *Mettl3* null cell divisions were largely unaltered and remained within the plane of the basal layer (*Figure 6F*). However, at P0, a 45 min pulse with EdU revealed signs of

elevated proliferation in the cKO basal cells (*Figure 6—figure supplement 2B*). This was surprising as pups did not grow in size appreciably post-birth (*Figure 2—figure supplement 2A*). As we show below, this could also not be accounted for by an appreciable rise in apoptosis. Rather, we attribute this change to reflect replacements due to the increased departure of basal cells into the suprabasal layers. This was accompanied by a marked increase in cellularity, particularly within the spinous layers (*Figure 6G*, *Figure 6—figure supplement 2B*). Suprabasal cells were also smaller than normal (*Figure 6—figure supplement 2C*). The increase of cell number and decrease of cell size in the suprabasal layers was also confirmed by FACS quantifications, in which we found the numbers of K10$^+$ cells were greater, with somewhat lower values on the forward scatter projection (*Figure 6—figure supplement 2D*).

Although the four stages of terminal differentiation in *Mettl3* cKO epidermis were still recognizable by morphology and by molecular markers (*Figure 6G*, *Figure 6—figure supplement 2E*), perturbations reverberated into the later stages of differentiation. In normal skin, when spinous cells transition to the granular layer, they initially remain transcriptionally active, but then a destructive phase ensues as they lose nuclei and other organelles and flatten to become dead 'squames' (*Quiroz et al., 2020*). This is normally accompanied by DNA destruction, detected by TUNEL staining (*Figure 6—figure supplement 2F*). In contrast, *Mettl3* cKO epidermis displayed an appreciable number of nuclei in their granular layers (*Figure 6G*) and showed little or no signs of TUNEL positive cells (*Figure 6—figure supplement 2F*).

Additionally, although desmosomes and hemidesmosomes were still present, small gaps were often noted ultrastructurally at intercellular borders in the *Mettl3* cKO epidermis (arrow in *Figure 6G*, middle frame). We also observed occasional signs of spinous cell cytolysis, which was also reflected in the increased incidence of early suprabasal cells positive for TUNEL but negative for activated Caspase-3 (*Figure 6—figure supplement 2F*). Given the many perturbations seen in the *Mettl3* null epidermis, it was remarkable that the skin barrier of neonatal mice was still functional as judged by its ability to retain fluids (*Figure 2—figure supplement 2C*).

## Signs of compensatory mechanisms and RNA metabolism revealed in the mRNAs that are upregulated upon METTL3 loss

Finally, we turned to the mRNAs that were significantly upregulated upon METTL3 loss. We began by examining the expression status of *Mettl3* cKO *versus* control E17 skin mRNAs whose coding sequence SN-uTPM per nt values from our miCLIP data had been among the top 20% of heavily m6A-modified transcripts. Interestingly, a considerable cohort surfaced whose mRNA levels were significantly upregulated upon m6A loss, and which had a higher dynamic range of fold-changes than the downregulated mRNAs (*Figure 7A*, red dots in the right quadrant; *Supplementary file 3*, *4* with a cut-off Z score >1.96). In fact, mRNAs that showed high m6A modifications within their coding sequences were generally more up- than downregulated upon METTL3 loss (*Figure 6—figure supplement 1* and *Figure 7—figure supplement 1A*). These findings were consistent with prior reports that m6A can promote RNA degradation (*Du et al., 2016*; *Wang et al., 2014a*; *Zaccara and Jaffrey, 2020*).

In all four populations, the major enriched GO term categories of highly modified mRNAs in wild-type that were significantly upregulated upon m6A loss revealed changes in various aspects of RNA metabolism, including RNA binding, RNA processing, RNA modification, ribosome and translation, as well as ribonucleoprotein complex (*Figure 7B*; *Supplementary file 5*). Examples of the mRNAs within specific categories (color coded by category) are shown in the t-SNE plots in *Figure 7C* and validations by qPCR are shown in *Figure 7D*. The categories of mRNAs enriched upon METTL3 loss and highly m6A-modified in wild-type conditions raised the possibility that the upregulated mRNAs might represent a global response to m6A loss.

Interestingly, CAP-independent translation was among the upregulated GO terms, consistent with the reduced mRNA levels that we had detected for canonical initiation. RNA methylation was also among the GO terms. Additionally, of note, mRNAs encoding all three of the major m6A reader proteins—*Ythdf1*, *Ythdf2* and *Ythdf3* (*Wang et al., 2015*; *Wang et al., 2014a*)—were not only heavily m6A-modified in their coding sequences (*Supplementary file 4*), but also significantly upregulated upon m6A loss (Z-scores ranging from 3.8 to 16) (*Supplementary file 3*). These findings were suggestive of a possible feedback mechanism to sense m6A paucity and compensate. The upregulation of Pelota (*Pelo*) was also intriguing (*Figure 7C*). Pelota is a component of the

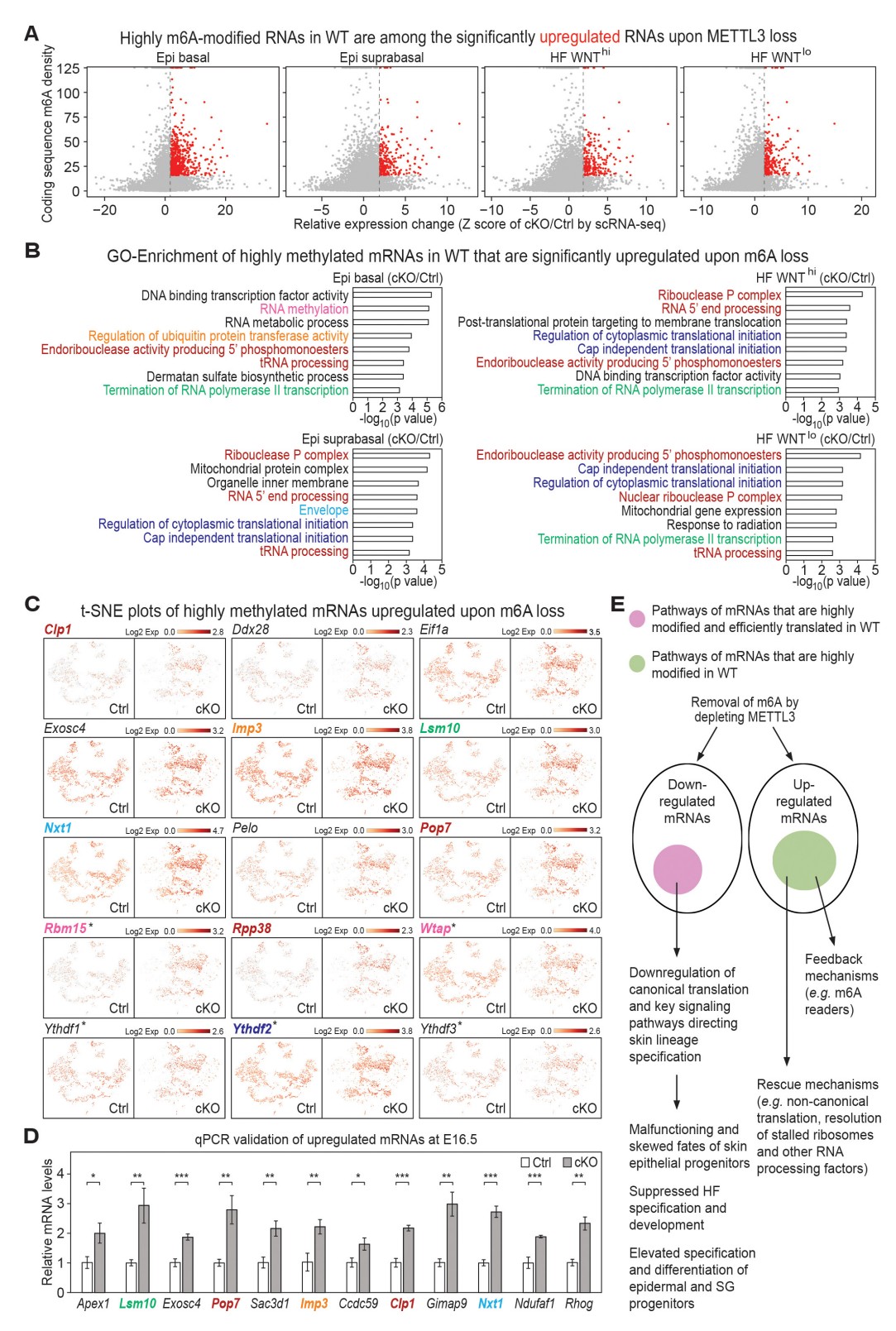

**Figure 7.** Investigation of RNAs upregulated upon METTL3 loss. (**A**) Scatter plots of mRNAs in the indicated groups of cells (as in *Figure 5A*) based on expression changes (cKO/Ctrl) assessed by scRNA-seq Z score and correlated with the coding sequence m6A density in wild-type (WT) skin epithelium assessed by the miCLIP SN-uTPM per nt value. Dots on the right of the dashed line in each plot indicate RNAs which from scRNA-seq have a Z score (cKO/Ctrl)>1.96 (*Supplementary file 3*). Among those, red dots denote mRNAs whose m6A coding sequence SN-uTPM per nt was among the top 20%

*Figure 7 continued on next page*

*Figure 7 continued*

(*Supplementary file 4*). (B) GSEA of transcripts with Z score (cKO/Ctrl)>1.96, FDR <0.05 in scRNA-seq and m6A coding sequence SN-uTPM per nt among the top 20% indicating the GO terms enriched. (C) Examination of the expression of selected upregulated mRNAs on the t-SNE plots. mRNA names are color-coded according to the GO terms they belong to in (B). Those mRNAs whose names are in black are known to be involved in translation regulation, and scored as elevated significantly upon METTL3 loss. *denotes mRNAs encoding factors related to m6A dynamics. (D) qPCR validation of increased levels of mRNAs in cKO/Ctrl. qPCR is on total RNA samples extracted from YFP⁺ skin epithelial cells FACS isolated from E16.5 embryos with *Tbp* mRNA as internal control (error bars: standard deviation, for each condition n = 3 biological replicates, *p<0.05 **p<0.01 ***p<0.001 by unpaired two-tailed Student's t-test). (E) Proposed model summarizing the effects of m6A loss on mRNA translation and degradation in the skin epithelia, and the consequences to their integrity and fate choices. The ovals represent all mRNAs that were either downregulated or upregulated significantly upon METTL3 loss. Pink circle: by first examining mRNAs that were heavily modified by m6A and also efficiently translated (*Figure 1*), and then independently identifying mRNAs that were among the most significantly downregulated upon METTL3 loss, we discovered considerable overlap in their pathways, suggestive of a translation block. The finding that a number of mRNAs involved in canonical translation were also significantly downregulated added to this notion. Factors involved in WNT signaling, NOTCH signaling and adhesion were featured prominently, in agreement with the morphogenetic defects observed. Green circle: these mRNAs were significantly upregulated upon METTL3 loss and were heavily m6A-modified in wild-type skin, but they did not correlate with translation efficiency. Rather, they encompassed mRNAs indicative of translational rescue pathways as well as feedback mechanisms.

The online version of this article includes the following source data and figure supplement(s) for figure 7:

**Source data 1.** qPCR of selected transcripts in (D).
**Figure supplement 1.** Additional analysis of features affected by the upregulated genes upon *Mettl3* cKO.
**Figure supplement 1—source data 1.** *Bmyc* qPCR in (B).
**Figure supplement 1—source data 2.** *Myc* qPCR in (C).
**Figure supplement 1—source data 3.** Quantification of MYC immunofluorescence signals in (D).

---

evolutionarily conserved quality control machinery that rescues stalled ribosomes, and it is also essential for skin epidermal integrity (*Liakath-Ali et al., 2018*).

Of additional note, MYC has long been linked to ribosomal biogenesis and translational control (*Piazzi et al., 2019*; *van Riggelen et al., 2010*), and MYC overexpression in embryonic epidermal progenitors is also known to favor epidermal and sebocyte differentiation at the expense of the HF lineage (*Berta et al., 2010*). In this regard, it is also noteworthy that both *Myc* and *Bmyc* mRNAs were not only highly modified by m6A within their coding sequences, but also by scRNA-seq, were significantly elevated within the *Mettl3* null epidermal progenitors of E17 skin (*Figure 7—figure supplement 1B,C*). Immunofluorescence microscopy was consistent with these differences (*Figure 7—figure supplement 1D*). Although modest, these increases followed the right trend to be a contributing factor to the *Mettl3* cKO phenotype.

## Discussion

Despite the increasing knowledge of the molecular mechanisms that underlie the roles of m6A modifications on mRNA degradation and translation, it has remained elusive how this abundant modification affects tissue biology. Our current study uncovered several new insights into m6A's actions and made headway in tying together and advancing our knowledge of certain recurrent themes that have emerged from prior in vivo and in vitro studies. As importantly, our findings place m6A as a key integrator of stem cell fate choices and translational control (*Figure 7E*).

In gaining new insights, we were aided by having previously performed in vivo ribosomal profiling on embryonic murine skin epithelium (*Sendoel et al., 2017*). By performing high-resolution miCLIP on this tissue, we were uniquely poised to tease out, on a global in vivo scale, a prominent correlation between the degree of m6A modifications that an mRNA has within its coding sequence and its translation efficiency. This correlation was particularly pronounced in mRNAs encoding signaling pathways known to affect skin progenitor fates. Moreover, by performing scRNA-seq on control and *Mettl3* cKO skin progenitors, we were further able to appreciate that many of the most significantly downregulated mRNAs upon m6A loss were associated with these same pathways.

The most striking phenotypic defects surfacing upon m6A loss were in HF morphogenesis, where both specification and down growth were markedly curtailed. These processes require WNT signaling, and mRNAs encoding a number of WNT regulators were significantly downregulated. Moreover, WNT-effectors nuclear LEF1 and β-catenin, were both diminished at the sites where WNT signaling is known to be essential. Further signs that WNT signaling was deleteriously affected

included the promotion of sebaceous gland fate characteristics at the expense of HF fate ones (*Merrill et al., 2001*; *Niemann et al., 2003*), the perturbations in SHH signaling and also the difficulty that *Mettl3* null HFs seemed to face in engulfing the DP.

Recent studies on other cell types and cancers have also reported connections between WNT signaling and m6A levels ( *Bai et al., 2019*; *Cui et al., 2020*; *Han et al., 2020*; *Liu et al., 2019*; *Miao et al., 2019*; *Tang et al., 2020*; *Zhang et al., 2019*). In some of these cases, m6A loss resulted in an elevation of WNT signaling, while in others, it was diminished, and a variety of mechanisms have been proposed to explain these seemingly context-dependent effects. Our findings show that not only are WNT pathway mRNAs among the most heavily m6A-modified and efficiently translated, but they are also among the most significantly downregulated early after m6A loss. When coupled with the physiological consequences to skin development, our findings place the impact of m6A alterations at the helm of this pathway.

For the skin, the link between WNT signaling and m6A was particularly intriguing because it offered a hitherto elusive connection between mRNA regulation and cellular fate choices. Additionally, given that overall levels of WNT signaling can profoundly impact fate specification, our studies reveal how even subtle changes in this pathway can have a profound impact on fate outcomes. That said, despite the importance of WNT signaling in skin progenitor fate specification and the consistency of the METTL3 loss of function phenotype with a decrease in this pathway, m6A modifications in the skin epithelium went well beyond this pathway. Indeed, our comprehensive analyses of the *Mettl3* cKO phenotype in the skin provided graphic appreciation that m6A RNA modifications function broadly and diversely in regulating cellular responses.

In addition to WNT signaling, the pathways affected when highly m6A-modified and efficiently translated mRNAs experienced depletion of m6A encompassed actin regulators, cell polarity, ECM-receptor interaction and NOTCH signaling. A number of cellular changes and phenotypic correlations were consistent with these differences. Thus for example, the perturbations we found in apical cytoskeletal polarity seem likely to contribute to the decreased ability of HF cells to engulf the DP. Additionally, the decrease in HES1 in early spinous cells was reflective of diminished NOTCH signaling and consistent with mRNA downregulation of NOTCH ligand JAG2 and NOTCH co-activator Mastermind like 2. And although we did not observe overt morphological differences in hemidesmosomes, the heightened departure of progenitors from the basal layer and their inability to form colonies in vitro were in agreement with possible defects in *Mettl3* null basal progenitor: ECM adhesion.

The high number of mRNAs upregulated significantly upon METTL3 loss were equally informative but in a strikingly different way. Although many of these transcripts were among the most highly m6A-modified in the wild-type state, their GO terms were not among the top m6A-modified mRNAs that were efficiently translated. On the contrary, the marked increase in their mRNA levels upon m6A ablation was suggestive that this mark normally functions to destabilize this cohort.

In contemplating the physiological relevance of the upregulated mRNAs in *Mettl3* cKO embryos, we were drawn to several curious possibilities. First, it was both intriguing and paradoxical to find that that *Myc* and *Bmyc* were upregulated upon m6A loss. In cancer cells, MYC has been implicated as a key positive regulator of CAP-dependent mRNA translation (*Cargnello and Topisirovic, 2019*), a process that was predicted to be downregulated upon *Mettl3* ablation in embryonc skin. That said, MYC overexpression has also been shown to promote differentiation in the epidermis and sebaceous glands at the expense of HF morphogenesis (*Arnold and Watt, 2001*; *Cottle et al., 2013*; *Watt et al., 2008b*), phenocopying some of the key features we observed upon m6A loss.

Notably, upregulation of RNA processing factors has been observed in the skin progenitors of both MYC-overexpressing skin (*Arnold and Watt, 2001*; *Frye et al., 2003*), and *Mettl3* cKO skin. As global rates in RNA metabolism can profoundly affect stem cells' activities as well as how they respond to stressful environments (*Blanco et al., 2016*; *Sampath et al., 2008*; *Sendoel et al., 2017*; *Signer et al., 2014*; *Starck et al., 2016*; *Zismanov et al., 2016*), it seems likely that the block in the degradation of this selected cohort of RNAs might impact the translation of the signaling pathways that orchestrate lineage specification.

A second major curiosity was that by miCLIP, *Ythdf1, Ythdf2* and *Ythdf3* fell within the top 20% of mRNAs whose coding sequence in basal skin progenitors was highly modified, and in *Mettl3* null basal progenitors, these mRNAs were among the most significantly increased in levels. YTHDF1-3 function as m6A readers. While YTHDF1 and 3 have been proposed to increase the stability and

translation of mRNAs (*Shi et al., 2017*; *Wang et al., 2015*), recent studies suggest that all three may function in localizing m6A-modified mRNAs to the decay machinery (*Wang et al., 2014a*; *Zaccara and Jaffrey, 2020*). Irrespective of their precise reader roles, their collective m6A modification in control progenitors and upregulation upon m6A loss is suggestive of a key rescue juncture for the cell.

Indeed many of the other upregulated pathways were also suggestive of potential feedback mechanisms, activated upon m6A loss. Among them were CAP-independent translation genes, RNA methylation genes, RNA processing pathways and ribosome-associated mRNA quality control genes. When coupled with the downregulation in mRNAs that are normally m6A-modified and highly translated, and with the reduction in canonical translation transcripts, these upregulated mRNA perturbations pointed not only to further signs of an altered protein translation state, but also to a potential built-in mechanism to boost alternative translational routes and RNA metabolism lost upon *Mettl3* ablation. Thus by having a cohort of key regulatory mRNAs that are heavily modified by m6A, progenitors are able to sense when the modification is lost and respond by rapidly enhancing their mRNA pool and calling into action rapid responses and rescue pathways to control the damage involved.

A model based on our findings and summarizing this speculation is presented in *Figure 7E*. Future studies will be needed to probe the significance of m6A modifications of individual mRNAs and to unearth the mechanisms underlying why some m6A-modified mRNAs were downregulated upon m6A loss while others were upregulated. It will also take further investigation to sift through the myriad of mRNAs and pathways that were affected both negatively and positively by m6A loss, and which exposed an underlying complexity in m6A's roles in translational regulation. The problem becomes even more daunting when considering that we have not yet examined the consequences of METTL3 loss to miRNAs in the skin. In this regard, in a breast cancer cell line, METTL3 was found to mark pri-miRNAs for recognition and processing (*Alarcón et al., 2015*), and in skin, METTL3 depletion shares certain similarities with global loss of miRNAs (*Yi et al., 2008*; *Yi and Fuchs, 2011*).

While many areas remain to be explored, our results show that the two arms of the pathways for down- and up-regulated mRNAs are intricately interwoven. In this regard, our many findings reported here shed new light on the previously identified roles of m6A in translational regulation. Finally, and intriguingly, the circuitry funneled to progenitor fate choices, each of which uses a distinct repertoire of signaling pathways and has distinct needs for protein synthesis. Our studies on m6A through the lens of skin morphogenesis now place m6A on the center stage of this arena.

## Materials and methods

### Mouse strains

The *Mettl3^fl* mouse strain was a gift from the Brüning Lab at Max Planck Institute for Metabolism Research and Policlinic for Endocrinology, Diabetes and Preventive Medicine. It was generated by inserting *LoxP* sites at both sides of the fourth exon of the *Mettl3* gene through homologous recombination with the constructed template from the Knockout Mouse Project Repository (*Cheng et al., 2019*). Sequences of primers for genotyping of the *Mettl3^fl* allele are listed in *Supplementary file 6*. The *Mettl3^fl* strain was then crossed with *Krt14-Cre*, *Rosa26-YFP^fl* and *Gli1-LacZ* strains to generate breeders. The *Gli1-LacZ* strain (Swiss Webster background) is from the Joyner Lab at Memorial Sloan Kettering Cancer Center. *Nude(Nu/Nu)* mice for grafts are from Charles Rivers Laboratories Strain 088. The wild-type mice for miCLIP and the *Mettl3^fl*, *Rosa26-YFP^fl*, *Krt14-Cre* mouse strains are under the C57BL/6 background. All animals were maintained in an American Association for the Accreditation of Laboratory Animal Care Internationally approved Comparative Bio-Science Center at the Rockefeller University and procedures were performed using Institutional Animal Care and Use Committee-approved protocols that ad-here to the standards of the National Institutes of Health.

### Back skin engraftment

Back skin dissected from P0 pups was incubated on a moist surface with phosphate-buffered saline (PBS) at 4°C overnight and then grafted to *Nu/Nu* female mice (6–8 weeks old) with the control and

cKO skin on each side of the back. After 15 days, bandages were removed, and the grafted skin pieces were harvested for analysis.

## EdU and BrdU labeling

For EdU pulse labeling at P0, EdU was applied on the pups at 25 µg per gram body weight through intraperitoneal injection at 45 min before sacrifice. For EdU chase-BrdU pulse, EdU was applied on pregnant female mice bearing E18.5 embryos at 25 µg per gram body weight through intraperitoneal injection, and BrdU was applied on the born P1 pups at 50 µg per gram body weight through intraperitoneal injection at 1 hr before sacrifice.

## Histology analysis, X-gal staining, Oil Red O staining and immunofluorescence on sagittal sections

For sagittal sections, back skin or whole embryos were embedded in OCT compound and cut into 10–14 µm sections on a Leica cryostat. The sections were mounted on SuperFrost Plus Adhesion slides (VWR), air-dried and fixed in PBS with 4% paraformaldehyde (PFA) at room temperature for 15 min.

Hematoxylin and eosin staining was performed with an adapted protocol from the University of California San Diego (http://mousepheno.ucsd.edu/hematoxylin.shtml).

X-gal staining was performed for intracellular β-galactosidase activity assessment. Slides were first stained with X-gal staining buffer (100 mM sodium phosphate buffer pH = 7.3, 1.3 mM $MgCl_2$, 3 mM $K_3Fe(CN)_6$, 3 mM $K_4Fe(CN)_6$, 1 mg/ml X-gal) at 37°C for 3 hr and then counter stained with nuclear fast red (Sigma-Aldrich, N3020) at room temperature for 5 min.

Oil Red O staining was performed in 2 µg/ml Oil Red O-isopropanol solution at room temperature for 1 hr followed with nuclei staining by hematoxylin.

For immunofluorescence, the slides were incubated in blocking buffer (1% fish gelatin, 5% normal donkey serum, 1% bovine serum albumin, 0.3% Triton X-100 in 1x PBS) at room temperature for 1 hr. Primary antibody staining was performed in the blocking buffer at 4°C overnight and secondary staining was performed in the blocking buffer at room temperature for 1 hr. Primary antibodies used: rat anti-BrdU (1:200, Abcam, ab6326); rat anti-CD104 (1:500, BD Pharmingen, 553745); armenian hamster anti-CD29 (1:500, BioLegend, 102201); rat anti-CD49f (1:1000, BioLegend, 313602); rabbit anti-cleaved Caspase-3 (1:200, R and D Systems, AF835); rabbit anti-FLG (1:2000, Fuchs lab); rabbit anti-iNV (1:2000, BioLegend, 924401); guinea pig anti-K5 (1:500, Fuchs lab); rabbit anti-K10 (1:1000, Covance, PRB-159P-100); guinea pig anti-LEF1 (1:5000, Fuchs lab); rabbit anti-LHX2 (1:5000, Fuchs lab); rabbit anti-LOR (1:4000, Fuchs lab); rabbit anti-METTL3 (1:500, Abcam, ab195352); rabbit anti-MYC (1:100, Abcam, ab32072); goat anti-PCAD (1:300, R and D Systems, AF761); rabbit anti-SOX9 (1:1000, Millipore, AB5535); rabbit anti-Survivin (1:500, Cell Signaling, 2808); chicken anti-GFP/YFP (1:1000, Abcam, ab13970). Secondary antibodies conjugated to Alexa Fluor 488, Rhodamine Red-X and Alexa Fluor 647 are from Life Technologies. In Situ Cell Death Detection Kit, TMR red (Sigma-Aldrich, Roche-12156792910) was used for TUNEL staining. AF647 Click-iT EdU cell proliferation kit for imaging (Thermo Fisher Scientific, C10340) was used for EdU labeling. AF647 Phalloidin (Thermo Fisher Scientific, A22287) was used for F-actin staining. Slides were mounted in ProLong Gold Antifade Mountant with DAPI (Thermo Fisher Scientific, P36941) and imaged under a Zeiss Axio Observer.Z1 epifluorescence microscope equipped with a Hamamatsu ORCA-ER camera (Hamamatsu Photonics), and with an ApoTome.2 (Zeiss) slider that reduces the light scatter in the fluorescent samples controlled by Zen software (Zeiss). Images were processed through Fiji (ImageJ).

For quantitative analyses, different groups of cells (epi basal, suprabasal, HF WNT[hi] and HF WNT[lo]) were identified according to PCAD staining. Average LEF1 and MYC immunofluorescence intensity was measured among cells identified in the same group from one image. Background was measured and subtracted for each channel. The fluorescence intensity was then normalized to the average values from the corresponding control samples to get relative signal values. Cell division angles were quantified as described in *Williams et al., 2011*. Generally, the division angle was between the spindle orientation determined by Survivin and DAPI staining and the basal membrane determined by integrin β4 staining. Other quantifications (HF numbers, EdU, BrdU, cell sizes) are based on counting/measuring specific features in the images. TUNEL and cleaved Caspase 3-

positive cells were quantified per HF (placode, germ and peg) and per cell number in the stratified layers, where positive cells were observed (basal, suprabasal layer one and late granular).

To reduce any bias in data collection, all data from each group were not analyzed until all images were collected. No statistical method was used to predetermined sample size, randomization and experiment blinding was not used. Each experiment was repeated with at least two replicates and data presented is from three or more embryos, same age. Significance of p value was set at <0.05. Statistical details for each experiment, including the statistical test used, the sample size for each experiment and p value can be found in the corresponding figure legend.

## Immunohistochemistry

Back skin dissected from neonates was fixed in PBS with 4% PFA at 4°C overnight. Then the tissue was washed in PBS with 0%, 35% and 70% EtOH gradually and sent to Histowiz for further processing. Generally, immunohistochemistry was performed on a Bond Rx autostainer (Leica Biosystems) with enzyme treatment (1:1000) using standard protocols. Antibodies used were rat monoclonal F4/80 primary antibody (1:200, eBioscience, 14–4801) and rabbit anti-rat secondary (1:100, Vector). Bond Polymer Refine Detection (Leica Biosystems) was used according to manufacturer's protocol. After staining, sections were dehydrated and film coverslipped using a TissueTek-Prisma and Coverslipper (Sakura). Whole slide scanning (40x) was performed on an Aperio AT2 (Leica Biosystems).

## Whole-mount immunofluorescence and confocal microscopy

For whole-mount immunofluorescence, embryos were fixed in PBS with 4% PFA for 1 hr, followed by extensive washing in PBS. Samples were then permeabilized for 3 hr in 0.3% Triton X-100 in PBS and treated with Gelatin Block (2.5% fish gelatin, 5% normal donkey serum, 3% BSA, 0.3% Triton, 1x PBS). The following primary antibodies were used: rabbit anti-ECAD (1:500, Cell Signaling, 3195); rabbit anti-HES1 (1:200; Fuchs lab); rabbit anti-METTL3 (1:100, Abcam, ab195352); goat anti-PCAD (1:600, R and D Systems, AF761); chicken anti-GFP/YFP (1:1200, Abcam, ab13970). Primary antibodies were incubated at 4°C for 36 hr. After washing with 0.1% Triton X-100 in PBS, samples were incubated overnight at 4°C with secondary antibodies conjugated to Alexa Fluor 488, Rhodamine Red-X and Alexa Fluor 647 (1:1000, Life Technologies). For E16.5 whole skin and P0 epidermis, samples were washed, counterstained with DAPI and mounted in SlowFade Diamond Antifade Mountant (Invitrogen). For P0 whole skin, samples were washed, counterstained with DAPI and processed though tissue clearing with ethyl cinnamate as described in *Gur-Cohen et al., 2019*. Confocal images of whole-mounts were acquired using a spinning disk confocal system (Andor Technology Ltd) equipped with an Andor Zyla 4.2 and a Yokogawa CSU-W1 (Yokogawa Electric, Tokyo) unit based on a Nikon TE2000-E inverted microscope. Four laser lines (405, 488, 561 and 625 nm) were used for near simultaneous excitation of DAPI, Alexa448, Rhodamine Red-X and Alexa647 fluorophores. The system was driven by Andor IQ3 software. 40x oil objective was used to acquire Z stacks of 0.5–1 μm steps.

$HES1^+$ cells were quantified using Fiji (ImageJ). Briefly, 40x Z stacks of spinning disk confocal images of inter follicular epidermis 2500 $μm^2$ regions were converted into composite images in which DAPI was in blue channel, YFP in the green channel and HES1 in the red channel. From each region, $HES1^+$ and $YFP^+$ cells were quantified per optical sections of basal, suprabasal 1 (S-1) and suprabasal 2 (S-2) layers. The numbers of $HES1^+$ cells fwere recorded and the proportions calculated relative to the total $YFP^+$ cells for each represented layer. A minimum of 25 regions were analyzed per embryo.

PCAD and ECAD immunofluorescence quantifications were performed using Fiji (ImageJ). Briefly, using spinning disk Z stacks of whole-mount 40x confocal images, we summed the intensity across the basal layer of 2500 $μm^2$ regions. The integrated density of PCAD and ECAD immuolabeling through the perimeter of the cell was measured and recorded for 10 cells in each region to a minimum of 18 regions per embryo. Background was then measured and subtracted for each channel. The ratio PCAD:ECAD was calculated per cell.

To reduce any bias in data collection, all data from each group were not analyzed until all images were collected. No statistical method was used to predetermined sample size, randomization and experiment blinding was not used. Each experiment was repeated with at least two replicates and data presented is from two or more embryos, same age. Distributions were tested for normality

using D'Agostino and Pearson test. To test significance, unpaired or paired two-tailed Student's t-tests were used for normal distribution and nonparametric Mann-Whitney test when the distribution did not follow a normal distribution. Significance of p value was set at <0.05. Statistical details for each experiment, including the statistical test used, the sample size for each experiment and p values can be found in the corresponding figure legend. All graphs and statistics were produced using GraphPad Prism 8.2 for MAC, GraphPad Software, San Diego, California, www.graphpad.com.

### Electron microscopy

Skin samples were fixed in 2% glutaraldehyde, 4% PFA, and 2 mM $CaCl_2$ in 0.05 M sodium cacodylate buffer, pH 7.2, at room temperature for over 1 hr. Then the samples were post-fixed in 1% osmium tetroxide, and processed for Epon embedding. Ultrathin sections (60 to 65 nm) were counterstained with uranyl acetate and lead citrate. Images were taken with a transmission electron microscope (Tecnai G2-12; FEI) equipped with a digital camera (AMT BioSprint29).

### Flow cytometry and cytospin analysis of skin epithelial cells

Preparation of embryonic and neonatal back skin for isolation or examination of epithelial cells by flow cytometry and staining protocols were performed as previously described (*Asare et al., 2017*; *Sendoel et al., 2017*). For E16.5 and E17 samples, back skin dissected from the embryos were treated with 0.25% collagenase and then 0.25% trypsin to get cell suspension. For P0 skin, back skin dissected from the neonates were treated with dispase first to separate the epidermis from dermis. Then the epidermis was digested with 0.25% trypsin to get cell suspension. Antibodies used: anti-CD29 PECy7 (1:1000, Thermo Fisher Scientific, 25–0291-82); anti-CD31 APC (1:1000, Thermo Fisher Scientific, 17–0311-82); anti CD45-APC (1:1000, Thermo Fisher Scientific, 17–0451-83); anti-CD49f PE (1:1000, Thermo Fisher Scientific, 12–0495-81); anti-CD49f PECy7 (1:1000, Thermo Fisher Scientific, 25–0495-82); anti CD117-APC (1:1000, Thermo Fisher Scientific, 17–1172-81); anti CD140a-APC (1:500, Thermo Fisher Scientific, 17–1401-81); rabbit anti-K10 (1:500, Biolegend, PRB-159P); mouse anti-rabbit IgG PE-Cy7 (1:300, Santa Cruz, sc-516721). CD31, CD45, CD117 and CD140a are lineage-negative markers. For FACS, dead cells were excluded by DAPI staining. The samples were proceeded to FACS on BD FACSAriaII sorter commended by the Diva software (BD Biosciences). For flow cytometry-based examination of K10[+] suprabasal cells, dead cells were excluded by LIVE/DEAD Fixable Aqua Dead Cell Stain Kit (Thermo Fisher Scientific, L34965). Antibody staining was performed after cells were fixed with Fixation/Permeabilization Solution Kit (BD Biosciences, 554714). Samples were analyzed on BD LSRII flow cytometer commended by the BD FACSDiva software (BD Biosciences) and data were processed with FlowJo. For cytospin analysis, cells were spin onto SuperFrost Plus Adhesion slides (VWR) with a Cytospin4 unit (Thermo/Shandon), and immunofluorescence staining was performed afterwards.

### Colony formation assay

YFP[+] epidermal cells were FACS isolated from E16.5 embryos and cell viability is examined by Trypan blue staining. 75,000 living cells were plated in each well of 6-well plates covered with mitomycin C-treated dermal fibroblasts. Cells were cultured in E-media supplemented with 15% (vol/vol) fetal bovine serum and 300 μM $Ca^{2+}$ in an incubator with 7% $CO_2$ in the air for 10 days. The cells were then fixed with 4% PFA in PBS and stained with chicken anti-GFP/YFP (1:1000, Abcam, ab13970) primary antibody and secondary antibody conjugated to Alexa Fluor 488. Image of each well was scanned with a Cytation 5 Cell Imaging Multi-Mode Reader (BioTek Instruments).

### Thin-layer chromatography

Measurement of m6A/A ratio in Poly(A)+ RNAs was performed as described in *Kruse et al., 2011*. YFP[+] epidermal cells were FACS isolated from control and cKO E16.5 embryos and collected in TRIzol-LS (Sigma-Aldrich, T3934) for total RNA extraction. Two rounds of Poly(A)+ RNA enrichment were performed with the Dynabeads mRNA Purification Kit (Thermo Fisher Scientific, 61006). 100 ng of the purified Poly(A)+ RNA was used for each biological replicate (pool of embryos with the same genotype) to perform the assay. The signal on the membrane was detected with a Typhoon Trio PhosphorImager (GE Healthcare) and quantified with Fiji (ImageJ).

## TEWL measurements

TEWL rate was assessed by Tewameter TM300 (Courage + Khazaka electronic GmbH as described in *Quiroz et al., 2020*). Basically neonates were sacrificed and their back skin was dissected and immediately spread over a clean, smooth surface. Over four TEWL measurements were collected on each piece of fully acclimatized skin. The values reported by the instrument were then normalized to the average values from the corresponding control samples to get relative water loss rate.

## Quantitative PCR

YFP$^+$ epidermal cells were FACS isolated from E16.5 embryos and collected in TRIzol-LS (Sigma-Aldrich, T3934). Total RNA were extracted with Direct-zol RNA Miniprep kit (Zymo Research, R2051) and treated with the RQ1 RNase-free DNase (Promega, M6101) to remove DNA contamination. cDNA was generated with the High-Capacity cDNA Reverse Transcription Kit (Thermo Fisher Scientific, 4368814) and qPCR was performed with SYBR green PCR Master Mix (Thermo Fisher Scientific, 4367660) on a QuantStudio 6 Flex Real-Time PCR System (Thermo Fisher Scientific), with *Tbp* mRNA as internal loading control. The numbers were further normalized to the average values from the corresponding control samples to get relative mRNA levels. Sequences of the primers are listed in *Supplementary file 6*.

## m6A individual-nucleotide-resolution cross-linking and immunoprecipitation

miCLIP was performed as described in *Grozhik et al., 2017*. Integrin α6-high epidermal cells were FACS isolated from wild-type P0 pups and collected in TRIzol-LS (Sigma-Aldrich, T3934) for total RNA extraction. Poly(A)+ RNA was then extracted with the Dynabeads mRNA Purification Kit (Thermo-Fisher, 61006) and treated with the RQ1 RNase-free DNase (Promega, M6101) to remove DNA contamination. 7–8 μg Poly(A)+ RNA was used for each biological replicate (pool of 3 litters of pups) to perform miCLIP. After fragmentation, 1/10 of the sample was saved as input to perform parallel library construction without CLIP. Libraries were sequenced on the Illumina Hi-seq platform to generate paired-ended 50 bp reads. Sequencing data was processed as described in *Grozhik et al., 2017*. Sequence composition nearby the m6A sites identified by CIMS analysis was analyzed with seqLogo/R package. Normalized-to-input uTPM over identified m6A sites (*Lawrence et al., 2013*) was counted using the GenomicRanges Bioconductor/R package. Basically a 21-nucleotide window centered at each identified m6A site was selected. The ratio between the total uTPM within the window counted from the miCLIP dataset and that from the corresponding input dataset was recorded as the normalized-to-input uTPM. The m6A sites were assigned to genes using the ChIPseeker package (*Yu et al., 2015*). All exon annotation as well as the nested coding sequence and UTR annotation of exons was extracted from our gene models in GTF format using the GenomicFeatures Bioconductor/R package (*Lawrence et al., 2013*). SN-uTPM for individual transcripts was then calculated as the sum of signal for all normalized-to-input uTPMs within the exons of a specific transcript. Visualization of m6A site distribution was performed as described in *Olarerin-George and Jaffrey, 2017*. Gene set enrichment analysis of m6A levels was performed using the fgsea Bioconductor package (*Sergushichev, 2016*) and the C2 (Curated pathways) and mSigDB C5 (GO)gene sets (*Subramanian et al., 2005*).

## Correlation analysis between the miCLIP and ribosome profiling data

The m6A abundance levels for transcripts were calculated as the sum of the normalized m6A counts within a gene's identified m6A sites over the normalized paired RNA-seq counts within these sites. Genes were binned into quintiles based on these normalized m6A abundance levels. Translational efficiencies were retrieved from the lab's prior in vivo ribosomal profiling of neonatal skin progenitors (*Sendoel et al., 2017*). The translational efficiency of the top and bottom 20% gene quintiles were compared visually by empirical cumulative distribution function (ECDF) plots and the significance of observed differences between these quintiles was assessed by Wilcoxon rank sum tests.

To capture putative functional enrichment for pathways in the genes which are both highly m6A-modified and have a high translational efficiency in wild-type cells, genes were binned again into quintiles by translational efficiency. Genes within the top 20% of translational efficiency and top 20%

normalized m6A modification level were tested for functional enrichment of KEGG pathways using the GOseq R/Bioconductor package.

For the comparison of translational efficiency in functional gene sets, gene set members were compared to all genes outside the functional set by both visual inspection in ECDF plots and statistical testing by Wilcoxon rank sum tests.

## Single-cell RNA sequencing

YFP[+] epidermal cells were FACS isolated from E17 embryos and cell viability was examined by Trypan blue staining. For both control and cKO samples, the ratio of living cells was >90%. 8000 of single cells from each sample was processed with the Chromium Single Cell 3' Library and Gel Bead Kit (V2) (10x Genomics, PN-120267) to prepare scRNA-seq libraries, which were then sequenced on Illumina NextSeq 500 sequenceras $26 \times 57 \times 8$.

The sequencing data were analyzed and aggregated using Cellranger (version 2.1.1) count and Aggr function with default setting, respectively. The aggregated datasets were processed by Seurat (version 2.3.4) (*Stuart et al., 2018*). The cells expressing less than 1800 genes and the genes expressed in less than 10 cells were removed from further consideration. Counts of genes in each cell were normalized and log10 transformed. The cell cycle phase was estimated by Seurat with default setting. And the mitochondrial transcripts in each cell were also calculated. Then, the data was rescaled to remove the effects of cell cycle and mitochondrial transcripts. The top 2000 variable genes were used to principle component analysis. The first 10 principle components were selected for clustering analysis. The clustering results were projected in t-SNE plots. The cell types were assigned according to the well-known marker gene expressions, including *Krt14, Krt15, Bmpr1b, Wnt3a, Fzd10, Krt1, Krt10, Sox9, Trps1, Shh, Lhx2, Dkk4, Fgfr1, Pthlh, Nfatc1, Tgfb2, Wif1, Krt17* and *Sox2.*

Pseudo-time estimation of lineage differentiation trajectories was performed with monocle (version 2.10.1) (*Qiu et al., 2017a*; *Qiu et al., 2017b*; *Trapnell et al., 2014*). In the beginning, the expression levels of *Dkk4, Shh* and *Lhx2* were applied to cell type differentiation. Then, the cell clustering, pseudo-time and trajectory estimation was based on the user manual of monocle (http://cole-trapnell-lab.github.io/monocle-release/docs/#constructing-single-cell-trajectories).

Differential gene expression analysis and single-cell GSEA were processed by MAST (version 1.8.2, *Finak et al., 2015*). The comparison between control and cKO in each cluster was processed by default setting with 50 times bootstrap. The genes with p<0.05 and pathways with false discovery rates <0.25 were selected.

Visualization of m6A levels against scRNA-seq Z scores was performed using the ggplot2 R libraries. Functional enrichment analyses of genes with high m6A (top 20%) and upregulated in *Mettl3* cKO conditions [Z score (cKO/Ctrl)>1.96] were performed using the weighted hypogeometric test in the GOseq Bioconductor package using the C5 and C2 mSigDB gene sets (*Subramanian et al., 2005*).

## Statistical analyses

All statistical analyses are provided for each of the individual methods sections. Additionally, statistical and graphical data analyses were performed using Microsoft Excel and Prism 8 (GraphPad) software. For measurements, ≥2 biological replicates and two or more technical replicates were used, where applicable. To determine the significance between two groups, comparisons were made using an unpaired two-tailed Student's t-test or analysis of variance, as appropriate. Multiple testing correction was done using the Benjamini–Hochberg method. In the box and whisker plots, the middle line represents the median, the upper and lower hinges correspond to the first and third quartiles, and the upper and lower whiskers display the full range of variation (minimum to maximum). Most experiments were repeated on ≥3 pairs of sample and control sets.

## Acknowledgements

We thank Dr Jens Brüning and Dr Martin E. Hess at Max Planck Institute for Metabolism Research and Policlinic for Endocrinology, Diabetes and Preventive Medicine, Cologne, Germany for sharing the *Mettl3[fl]* mouse strain. We thank Rockefeller University's Flow Cytometry Resource Center (Svetlana Mazel, director), the Genomics Resource Center (Connie Zhao, director) and the American

Association for the Accreditation of Laboratory Animal Care-accredited comparative biology center (R Tolwani, director) for their services. We thank the Epigenomics Core Facility (Yushan Li, director) at Weill Cornell University for sample processing. We also thank members of the Fuchs lab, specifically John Levorse, Brian Hurwitz, Stephanie Ellis, Katherine S Stewart, Matthew Tierney, Felipe Garcia Quiroz, Vincent F Fiore, Shaopeng Yuan, Rachel Niec, Nicholas Gomez, Amma Asare, Aaron Mertz, Sanjee Baksh, Hanseul Yang, Melanie Laurin, Yejing Ge, Maria Nikolova, Ellen Wong, June Racelis, Megan Sribour and Lynette Hidalgo as well as those of the Jaffrey lab, specifically Anya V Grozhik, Brian Pickering, Sara Zacara, Anthony O Olarerin-George, Jan Mauer and Pierre Klein for thorough discussions and for assistance in setting up the various experiments performed in this study. Dr. Robert Roeder's lab at Rockefeller University also offered us valuable technical support in setting up the miCLIP experiment. LX is the Dale F and Betty Ann Frey Fellow of the Damon Runyon Cancer Research Foundation, DRG-2263–16. EF is an investigator of the Howard Hughes Medical Institute. The work is also supported by grants from the National Institutes of Health (R01-AR27883 to EF, R01-CA186702 and R21-CA224391 to SRJ)

## Additional information

### Competing interests

Elaine Fuchs: Reviewing editor, *eLife*. Samie R Jaffrey: Scientific founder, advisor to, and owns equity in Gotham Therapeutics. The other authors declare that no competing interests exist.

### Funding

| Funder | Grant reference number | Author |
| --- | --- | --- |
| National Institutes of Health | R01-AR27883 | Elaine Fuchs |
| National Institutes of Health | R01-CA186702 | Samie R Jaffrey |
| National Institutes of Health | R21-CA224391 | Samie R Jaffrey |
| Damon Runyon Cancer Research Foundation | DRG-2263-16 | Linghe Xi |
| HHMI | | Elaine Fuchs |

The funders had no role in study design, data collection and interpretation, or the decision to submit the work for publication.

### Author contributions

Linghe Xi, Conceptualization, Data curation, Formal analysis, Funding acquisition, Validation, Investigation, Visualization, Methodology, Writing - original draft, Project administration, Writing - review and editing; Thomas Carroll, Conceptualization, Data curation, Software, Formal analysis, Validation, Visualization, Methodology, Writing - review and editing; Irina Matos, Conceptualization, Data curation, Formal analysis, Validation, Investigation, Visualization, Methodology, Writing - review and editing; Ji-Dung Luo, Data curation, Software, Formal analysis, Visualization, Methodology; Lisa Polak, Investigation, Methodology, Acquisition of data and intellectual input; H Amalia Pasolli, Investigation, Visualization, Methodology, Data Analysis and Interpretation; Samie R Jaffrey, Conceptualization, Resources, Supervision, Writing - review and editing; Elaine Fuchs, Conceptualization, Supervision, Funding acquisition, Methodology, Writing - original draft, Project administration, Writing - review and editing

### Author ORCIDs

Linghe Xi ⓘD https://orcid.org/0000-0002-1239-5316
Irina Matos ⓘD http://orcid.org/0000-0001-6100-8020
Ji-Dung Luo ⓘD http://orcid.org/0000-0003-0150-1440
Samie R Jaffrey ⓘD https://orcid.org/0000-0003-3615-6958
Elaine Fuchs ⓘD https://orcid.org/0000-0002-0978-5137

## Ethics

Animal experimentation: All mouse strains were housed in an AAALAC-accredited facility and experiments were conducted according to the Rockefeller University's Institutional Animal Care and Use Committee (IACUC), and NIH guidelines for Animal Care and Use. All animal procedures used in this study are described in our #20012-H & #17091-H protocols, which have been previously reviewed and approved by the Rockefeller University IACUC.

## Decision letter and Author response

Decision letter https://doi.org/10.7554/eLife.56980.sa1
Author response https://doi.org/10.7554/eLife.56980.sa2

# Additional files

## Supplementary files

• Supplementary file 1. Summary of all identified m6A sites through miCLIP.

• Supplementary file 2. Quantification of m6A levels based on the sum of normalized-to-input uTPM value of m6A along coding sequence (CDS SN-uTPM) and GSEA. First sheet: Rank of mRNAs based on coding sequence SN-uTPM. Second sheet: GSEA of mRNAs weighted on coding sequence SN-uTPM. The gene sets with p values <0.25 are shown. Third sheet: GSEA of mRNAs with top 20% coding sequence SN-uTPM and top 20% translation efficiency. The gene sets with p values <0.10 are shown.

• Supplementary file 3. Differential gene expression analysis through scRNA-seq. The extent of differential gene expression assessed by Z score (reflecting the extent of differential expression) and false discovery rate (FDR) was calculated between groups of Ctrl and cKO cells with the same identity, as indicated by sheet names in the file.

• Supplementary file 4. Different parameters used to assess m6A modification levels.

• Supplementary file 5. GSEA of transcripts with Z score (cKO/Ctrl)>1.96, FDR <0.05 in scRNA-seq and m6A coding sequence SN-uTPM per nt among the top 20%. The gene sets with p values <0.05 are shown.

• Supplementary file 6. Sequences of genotyping and qPCR primers used in this study.

• Transparent reporting form

## Data availability

The miCLIP and scRNA-seq data that support the findings of this study have been deposited to the Gene Expression Omnibus (GEO) repository with the accession codes GSE147415, GSE147489, and GSE14749.

The following datasets were generated:

| Author(s) | Year | Dataset title | Dataset URL | Database and Identifier |
|---|---|---|---|---|
| Xi L, Fuchs E | 2020 | Single-cell RNA-seq of embryonic day 17 (E17) mouse skin epithelial cells with or without Mettl3 knockout | https://www.ncbi.nlm.nih.gov/geo/query/acc.cgi?acc=GSE147415 | NCBI Gene Expression Omnibus, GSE147415 |
| Xi L, Fuchs E | 2020 | miCLIP-seq of postnatal day 0 (P0) normal mouse skin epithelial cells | https://www.ncbi.nlm.nih.gov/geo/query/acc.cgi?acc=GSE147489 | NCBI Gene Expression Omnibus, GSE147489 |
| Xi L, Fuchs E | 2020 | mouse skin epithelial cells | https://www.ncbi.nlm.nih.gov/geo/query/acc.cgi?acc=GSE147490 | NCBI Gene Expression Omnibus, GSE147490 |

The following previously published dataset was used:

| | | | | Database and |
|---|---|---|---|---|

| Author(s) | Year | Dataset title | Dataset URL | Identifier |
|---|---|---|---|---|
| Sendoel A, Fuchs E | 2017 | Epidermis-specific ribosome profiling to describe the translational landscape of SOX2 | https://www.ncbi.nlm.nih.gov/geo/query/acc.cgi?acc=GSE83332 | NCBI Gene Expression Omnibus, GSE83332 |

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

# Appendix 1

**Appendix 1—key resources table**

| Reagent type (species) or resource | Designation | Source or reference | Identifiers | Additional information |
|---|---|---|---|---|
| Genetic reagent (*M. musculus*) | C57BL/6J | Jackson lab | Stock #: 000664 | |
| Genetic reagent (*M. musculus*) | *Mettl3*<sup>flox</sup> | PMID:31412241 J. Brüning (Max Planck Institute for Metabolism Research and Policlinic for Endocrinology, Diabetes and Preventive Medicine) | | |
| Genetic reagent (*M. musculus*) | *Krt14-Cre* | PMID:10411913 E. Fuchs (Rockefeller University) | MGI:1926500 | |
| Genetic reagent (*M. musculus*) | *Rosa26-YFP*<sup>flox</sup> | Jackson lab | Stock #: 006148 | |
| Genetic reagent (*M. musculus*) | *Gli-LacZ* | PMID:12361967 A. Joyner (MSKCC) | MGI:2449767 | |
| Genetic reagent (*M. musculus*) | *Nude* (*Nu/Nu*) | Charles River Laboratories | Strain code: 088 | |
| Antibody | Rat anti-BrdU | Abcam | Cat. #: ab6326 RRID:AB_305426 | IMF (1:200) |
| Antibody | Rat anti-CD104 | BD Pharmingen | Cat. #: 553745 RRID: AB_395027 | IMF (1:500) |
| Antibody | Armenian hamster anti-CD29 | BioLegend | Cat. #: 102201 RRID:AB_312878 | IMF (1:500) |
| Antibody | Rat anti-CD49f | BioLegend | Cat. #: 313602 RRID:AB_345296 | IMF (1:1000) |
| Antibody | Rabbit anti-Cleaved Caspase-3 | R&D Systems | Cat. #: AF835 RRID:AB_2243952 | IMF (1:200) |
| Antibody | Rabbit anti-ECAD | Cell Signaling | Cat. #: 3195 RRID:AB_2291471 | IMF (1:500) |
| Antibody | Rabbit anti-FLG | E. Fuchs (Rockefeller University) | | IMF (1:2000) |
| Antibody | Rabbit anti-HES1 | E. Fuchs (Rockefeller University) | | IMF (1:200) |
| Antibody | Rabbit anti-iNV | BioLegend | Cat. #: 924401 RRID:AB_2565452 | IMF (1:2000) |
| Antibody | Guinea pig anti-K5 | E. Fuchs (Rockefeller University) | | IMF (1:500) |
| Antibody | Rabbit anti-K10 | Covance | Cat. #: PRB-159P-100 RRID:AB_291580 | IMF (1:1000) Flow (1:500) |

*Continued on next page*

*Appendix 1—key resources table continued*

| Reagent type (species) or resource | Designation | Source or reference | Identifiers | Additional information |
|---|---|---|---|---|
| Antibody | Guinea pig anti-LEF1 | E. Fuchs (Rockefeller University) | | IMF (1:5000) |
| Antibody | Rabbit anti-LHX2 | E. Fuchs (Rockefeller University) | | IMF (1:5000) |
| Antibody | Rabbit anti-LOR | E. Fuchs (Rockefeller University) | | IMF (1:4000) |
| Antibody | Rabbit anti-METTL3 | Abcam | Cat. #: ab195352 RRID:AB_2721254 | IMF (1:100-1:500) |
| Antibody | Rabbit anti-MYC | Abcam | Cat. #: ab32072 RRID:AB_731658 | IMF (1:100) |
| Antibody | Goat anti-PCAD | R&D Systems | Cat. #: AF761 RRID:AB_355581 | IMF (1:300-1:600) |
| Antibody | Rabbit anti-SOX9 | Millipore | Cat. #: AB5535 RRID:AB_2239761 | IMF (1:1000) |
| Antibody | Rabbit anti-Survivin | Cell Signalling | Cat. #: 2808 RRID:AB_2063948 | IMF (1:500) |
| Antibody | Chicken anti-GFP/YFP | Abcam | Cat. #: ab13970 RRID:AB_300798 | IMF (1:1000-1:1200) |
| Antibody | anti-CD29 PECy7 | Thermo Fisher Scientific | Cat. #: 25-0291-82 RRID: AB_1234962 | FACS (1:1000) |
| Antibody | anti-CD31 APC | Thermo Fisher Scientific | Cat. #: 17-0311-82 RRID:AB_657735 | FACS (1:1000) |
| Antibody | anti CD45-APC | Thermo Fisher Scientific | Cat. #: 17-0451-83 RRID:AB_469393 | FACS (1:1000) |
| Antibody | anti-CD49f PE | Thermo Fisher Scientific | Cat. #: 12-0495-81 RRID:AB_891478 | FACS (1:1000) |
| Antibody | anti-CD49f PECy7 | Thermo Fisher Scientific | Cat. #: 25-0495-82 RRID:AB_10804881 | FACS (1:1000) |
| Antibody | anti CD117-APC | Thermo Fisher Scientific | Cat. #: 17-1172-81 RRID:AB_469432 | FACS (1:1000) |
| Antibody | anti CD140a-APC | Thermo Fisher Scientific | Cat. #: 17-1401-81 RRID:AB_529482 | FACS (1:500) |
| Antibody | Mouse anti-rabbit IgG PE-Cy7 | Santa Cruz | Cat. #: sc-516721 | Flow (1:300) |
| Chemical compound, drug | Alexa Fluor 647 Phalloidin | Thermo Fisher Scientific | Cat. #: A22287 RRID:AB_2620155 | IMF (1:50) |
| Commercial assay or kit | In situ cell death detection kit, TMR red | Sigma-Aldrich | Cat. #: 12156792910 | |
| Commercial assay or kit | Click-iT EdU cell proliferation kit for imaging, Alexa Fluor 647 dye | Thermo Fisher Scientific | Cat. #: C10340 | |
| Commercial assay or kit | LIVE/DEADfixable Aqua dead cell stain kit, for 405 nm excitation | Thermo Fisher Scientific | Cat. #: L34957 | |
| Commercial assay or kit | Fixation/ permeabilization solution kit | BD Biosciences | Cat. #: 554714 | |
| Commercial assay or kit | Dynabeads mRNA purification kit | Thermo Fisher Scientific | Cat. #: 61006 | |
| Commercial assay or kit | Direct-zol RNA miniprep kit | Zymo Research | Cat. #: R2051 | |

*Continued on next page*

*Appendix 1—key resources table continued*

| Reagent type (species) or resource | Designation | Source or reference | Identifiers | Additional information |
|---|---|---|---|---|
| Commercial assay or kit | RQ1 RNase-free DNase | Promega | Cat. #: M6101 | |
| Commercial assay or kit | High-capacity cDNA reverse transcription kit | Thermo Fisher Scientific | Cat. #: 4368814 | |
| Commercial assay or kit | Power SYBR Green PCR Master Mix | Thermo Fisher Scientific | Cat. #: 4367660 | |
| Commercial assay or kit | Chromium single cell 3' reagent kit | 10x Genomics | Cat. #: PN-120267 | |
| Chemical compound, drug | TRI Reagent LS | Sigma-Aldrich | Cat. #: T3934 | |
| Software, algorithm | Zen | Zeiss | https://www.zeiss.com/microscopy/int/products/microscope-software/zen.html | |
| Software, algorithm | Fiji ImageJ | ImageJ | https://imagej.net/Fiji | |
| Software, algorithm | Andor IQ3 | Oxford Instruments | https://andor.oxinst.com/products/iq-live-cell-imaging-software/ | |
| Software, algorithm | GraphPad Prism 8.2 | GraphPad | https://www.graphpad.com/ | |
| Software, algorithm | BD FACSDiva | BD Biosciences | https://www.bdbiosciences.com/en-us/instruments/research-instruments/research-software/flow-cytometry-acquisition/facsdiva-software | |
| Software, algorithm | FlowJo | FlowJo, LLC | https://www.flowjo.com/solutions/flowjo | Version 10 |
| Software, algorithm | R | CRAN | https://cran.r-project.org | R Version 3.6.3 Bioconductor Version 3.10 |
| Software, algorithm | Flexbar | PMID:24832523 | https://github.com/seqan/flexbar | Version 2.5 |
| Software, algorithm | pyCRAC | S. Granneman (SynthSys) | http://sandergranneman.bio.ed.ac.uk/pycrac-software | Version 1.1.3 |
| Software, algorithm | Novoalign | Novocraft | http://www.novocraft.com/products/novoalign/ | Version 3.04.06 |

*Continued on next page*

*Appendix 1—key resources table continued*

| Reagent type (species) or resource | Designation | Source or reference | Identifiers | Additional information |
|---|---|---|---|---|
| Software, algorithm | CIMS | C. Zhang (Columbia University) | https://zhanglab. c2b2.columbia. edu/index.php/ CTK_ Documentation | |
| Software, algorithm | Bedtools | PMID:20110278 A. Quinlan (University of Utah) | https://github.com/ arq5x/bedtools2.git | Version 2.25.0 |
| Software, algorithm | Samtools | PMID:19505943 H. Li (Harvard University) | http://samtools. sourceforge.net | Version 1.2 |
| Software, algorithm | Cell Ranger | 10x Genomics | https://support. 10xgenomics.com/ single-cell-gene-expression/ software/overview/ welcome | Version 2.1.1 |
| Software, algorithm | Seurat | PMID:29608179 R. Satija (New York University) | https://satijalab.org/ seurat/ | Version 2.3.4 |
| Software, algorithm | Monocle | PMID:28114287 C. Trapnell (University of Washington) | http://cole-trapnell-lab. github.io/monocle-release/ | Version 2.10.1 |
| Software, algorithm | MAST | PMID:26653891 A. McDavid (University of Rochester Medical Center) | https://www. bioconductor.org/ packages/release/ bioc/html/MAST. html | Version 1.8.2 |

