## [Decision Letter]

**Acceptance summary:**

Overall, this manuscript provides informative data sets of m6A profiles and clear-cut evidence showing the consequences of m6A loss in skin epidermal progenitors. This study highlights the imperative role of m6A modification in modulating fate decision of skin epidermal progenitors.

**Decision letter after peer review:**

Thank you for submitting your article "m6A impacts fate choices during skin morphogenesis" for consideration by *eLife*. Your article has been reviewed by Marianne Bronner as the Senior Editor, a Reviewing Editor, and two reviewers. The following individuals involved in review of your submission have agreed to reveal their identity: Rui Yi (Reviewer #1).

The reviewers have discussed the reviews with one another and the Reviewing Editor has drafted this decision to help you prepare a revised submission.

Summary:

In this manuscript, Xi et al., studied m6A RNA modification and its role in embryonic skin development. They first profiled m6A modification in epithelial progenitors and investigated the correlation between m6A modification and translation efficiency as well as mRNA stability. By conditionally deleting *Mettl3*, a critical writer for m6A, in epithelial cells during skin development, they examined the physiological impact of *Mettl3*/m6A modification and observed compromised hair follicle morphogenesis.

Essential revisions:

1) The correlation between m6A and translation efficiency (TE) seems weak, and the way they show this in Figure 1E is confusing. In the Materials and methods section, I couldn't find the definition of "Relative TE" and the cut-off for "high", "med" and "low" m6A modification. As it's shown in Figure 1E, ~50% of "high" m6A mRNA have relative TE greater than 0 and ~50% less than 0. This doesn't support "high" m6A enhances TE. Even in the CDS panel, "med" and "low" m6A transcripts show only very mild reduction of overall TE. They should fully define the parameters used in this analysis; use non-modified transcripts as control; calculate the statistical significance of each comparison e.g. "high" vs "med", "med" vs "low" and all against non-modified transcripts. To truly support the claim that m6A enhances TE, they should perform ribosome profiling in WT and mettl3 ko epithelial cells and calculate the differential TE. And for a few examples, they should quantify protein levels by Western blotting and/or staining to support the increased TE indeed leads to a higher concentration of protein or the reduced protein levels in cKO.

2) They relied heavily on gene ontology analysis to identify pathways that are preferentially affected by m6A. However, this analysis is confusing. They stated they "ranked the mRNAs carrying the modification according to the sum of normalized-to-input uTPM at each m6A site within the coding sequence, and divided by CDS length" (subsection “Skin transcripts most highly modified by m6A are involved in hair follicle 129 morphogenesis”). However, I'm not sure if they used the ranking system at all when they searched for enriched KEGG pathways. It seems that they simply identified pathways with most transcripts with m6A but not necessarily highly modified. They need to show (1) which pathways the most highly m6A modified mRNAs are enriched in? (2) whether the top enriched pathways contain many highly m6A modified mRNAs? In addition, they should apply their TE analysis to these enriched pathways. Can they find reduced TE in these highly modified mRNAs that belong to the top KEGG pathways such as BCC and hedgehog signaling?

3) The phenotypical analysis should be strengthened by more careful analysis of cell proliferation such as colony formation in addition to EdU and Ki67 staining and cell adhesion analysis. The demonstrated phenotypes also seem similar to Dicer1 KO, when all miRNAs are depleted, including neonatal lethality due to the lack of weight gain, degenerated hair follicles and diminished filiform papillae formation.

4) They mentioned that WNT and SHH pathways surfaced in multiple pathways in their miCLIP data and suggested that m6A might function in hair follicle fate through these pathways. However, the authors did not show how these pathways behaved in WT vs cKO mice. Such analysis will strengthen their claims. Aberrant LEF and SHH signaling was associated with failure to form tight dermal condensates. With this data, they concluded that WNT signaling has been altered by m6A modification. However, there were not mechanistic studies to prove that WNT has been altered. The authors can do qPCR analysis of transcripts that are under WNT regulation to further show altered WNT signaling (e.g. axin) or IHC imaging of β-catenin etc. to future strengthen their conclusion.

5) They performed single-cell RNA-seq to measure the changed transcriptome and cellular state. However, their cell clustering analysis should be performed by using control and cKO samples together, rather than clustering separately and relying on marker genes to identify each population in control and cKO (Figure 5—figure supplement 1D, F). Are all corresponding cell populations in control and cKO clustered together? If not, they should explore the underlying reasons. The pseudotime analysis in Figure 4B should be strengthened by statistical analysis rather than eyeballing the population/lineage differences between Ctrl and cKO. They also should keep in mind they only have 1 pair of biological samples, which may reduce the robustness of the analysis. For differential gene analysis, they should independently confirm each data (Figure 4C, Figure 5D etc.) by qPCR or bulk RNA-seq from more sample pairs.

6) Based on ribosome profiling analysis, they suggest that m6A enhances translation. Based on RNA-seq, they suggest that m6A reduces mRNA stability. Globally, these are two opposite effects on mRNA's potential to make protein. It's important to distinguish (1) whether these two opposite effects of m6A are on different mRNAs? (2) if the same mRNAs are under the control of both effects, what is the net effect on protein synthesis? Are they cancelled out or one effect is more dominant than the other? How does this explain/correlate with the phenotypes?

7) The observation of prematurely detached basal cells is interesting. Are they the direct consequence of reduced cell-cell adhesion and/or cell-bm adhesion?

8) The reviewers were not sure about the point of OPP experiments. OPP label may be too low resolution to show any changes in *Mettl3* cKO.

9) Figure 3E-F: it is clear that in the grafted skin *Mettl3* cKO HFs underwent HF degeneration in accompany with sebocyte differentiation. However, it is not clear if this fate switch only occurred in *Mettl3* cKO HFs upon wounding (grafting in this case). Oil red O staining with P6 ungrafted skin sections will partially address this question. If this fate switch only happens upon wounding, authors should discuss the effects of m6A loss in the differential circumstance as well as speculate the effect of wounding in m6A-mediated regulation.

10) Subsection “Conditional ablation of *Mettl3* in epidermal progenitors results in a marked defect in HF morphogenesis”: "the engrafted cKO epidermis was hyperthickened, a feature which had also been evident in ungrafted cKO epidermis." To this reviewer, the hyperthicken epidermis could NOT be appreciated in ungrafted cKO epidermis based on the H&E staining shown in Figure 2G. The quantification in the thickness of ungrafted and grafted epidermis will be essential. In line with point #2, this hyperthicken epidermis in grafted cKO skin might be caused by wounding if hyperthickness is not detected in ungrafted *Mettl3* cKO epidermis.

11) Figure 5E: Authors stated that MYC expression was elevated within *Mettl3* cKO epidermal progenitors (subsection “A cohort of highly m6A-decorated mRNAs whose levels rise upon removal of their modification”). It is not clear to this reviewer how the quantification of MYC expression was done. Which cell markers did authors use to define "epidermal progenitors" (e.g. integrin α6 or K14…)? Did authors also include P-cad+ WNY-hi, WNT-lo HF cells, and epidermal suprabasal cells? Please clarify this by adding information in the figure legend as well as Materials and methods.

12) Figure 3—figure supplement 1: It is interesting to see high levels of PCAD expression in the basal layer of grafted *Mettl3* cKO epidermis. Did authors also find this striking pattern in ungrafted *Mettl3* cKO epidermis (e.g. P6 ungrafted skin)? What could be the explanation of this significant upregulation of P-Cadherin in the *Mettl3* cKO epidermal basal cells, e.g. alteration in cell junction? Confusion in fate specification? It will be helpful if authors can check on ungrafted skin and speculate the possibilities.

---

## [Author Response]

Essential revisions:1) The correlation between m6A and translation efficiency (TE) seems weak, and the way they show this in Figure 1E is confusing. In the Materials and methods section, I couldn't find the definition of "Relative TE" and the cut-off for "high", "med" and "low" m6A modification. As it's shown in Figure 1E, ~50% of "high" m6A mRNA have relative TE greater than 0 and ~50% less than 0. This doesn't support "high" m6A enhances TE. Even in the CDS panel, "med" and "low" m6A transcripts show only very mild reduction of overall TE. They should fully define the parameters used in this analysis; use non-modified transcripts as control; calculate the statistical significance of each comparison e.g. "high" vs "med", "med" vs "low" and all against non-modified transcripts.

We thank the reviewers for these comments and agree completely! Specifically, we have now added a schematic in new Figure 1F to show how the ribosomal and miCLIP analyses were performed on a population of neonatal skin progenitors. In response to the excellent reviewer comment, we also reanalyzed the data, this time taking the miCLIP-measured top 20% and bottom 20% of m6A-modified mRNAs across (1) full-length mRNA, (2) 5’UTR mRNA, (3) coding sequence mRNA and (4) 3’UTR mRNA, and then analyzing them for their relative translational efficiency as measured by ribosomal profiling. All the words are now spelled out to make it easy for the reader to see.

We have also performed the appropriate statistical tests to evaluate shifts in translation efficiency for our newly defined m6A abundance bins. Visualization of the empirical cumulative distribution function (ECDF) of the translation efficiency of these top 20% and bottom 20% quintiles was used to illustrate the correlation of m6A modification level with translational efficiency. To evaluate the statistical significance of these m6A dependent shifts in translation efficiency we have performed Mann-Whitney tests between our successive quintiles. This new analysis confirms and extends our earlier observation that transcripts with high m6A modification levels are more efficiently translated. All these new analyses are currently demonstrated in Figure 1 and Figure 1—figure supplement 1 and described in detail in the Materials and methods section.

To truly support the claim that m6A enhances TE, they should perform ribosome profiling in WT and mettl3 ko epithelial cells and calculate the differential TE.

We discussed this point in our initial response to the reviewers, and our response was accepted and approved by the editors/reviewers back in April. We actually had already tried, but technical hurdles beyond our control prevented us from performing ribosomal profiling on *Mettl3* KO epithelial cells either in vitro or in vivo. in vitro, the cells do not survive loss of *Mettl3*. Even when we culture the cells and then KO *Mettl3* in culture, the cells do not tolerate LOF. in vivo, the HF defects result in the placodes and germs fractionating with the dispase-treated dermis and not the epidermis at E17.5 and as development proceeds, the reduced HFs makes the fractionation unequal between KO and control. We now mention these issues in the text and provide the problems (new Figure 5—figure supplement 1A-C) so that other readers will know.

And for a few examples, they should quantify protein levels by Western blotting and/or staining to support the increased TE indeed leads to a higher concentration of protein or the reduced protein levels in cKO.

The inability to obtain sufficient numbers of matched cKO and Control cells (see above) precluded us from not only ribosomal profiling but also immunoblotting. However, as suggested, we have been successful at immunostaining and now add LEF1, β-catenin and *Gli1-lacZ* for our WNT and SHH analyses, and we provide closeups to show that the HF-DP borders are affected in ways that are expected when WNT signaling is inhibited (Matos et al., 2020). The data are now shown in new Figure 3A-D. By focusing on the top 20% of modified genes that are efficiently translated, the new comparisons also unearthed NOTCH signaling genes as ones highly modified and efficiently translated. We add data on NOTCH but later in the manuscript (new Figure 6D), when we focus on the epidermis. Equally importantly, by examining how the 20% most heavily modified mRNAs fare in scRNA seq, we uncover hitherto unappreciated strong ties between the data in Figure 1 and the single-cell *Mettl3* LOF data (new Figure 6).

2) They relied heavily on gene ontology analysis to identify pathways that are preferentially affected by m6A. However, this analysis is confusing. They stated they "ranked the mRNAs carrying the modification according to the sum of normalized-to-input uTPM at each m6A site within the coding sequence, and divided by CDS length" (subsection “Skin transcripts most highly modified by m6A are involved in hair follicle morphogenesis”). However, I'm not sure if they used the ranking system at all when they searched for enriched KEGG pathways. It seems that they simply identified pathways with most transcripts with m6A but not necessarily highly modified. They need to show (1) which pathways the most highly m6A modified mRNAs are enriched in? (2) whether the top enriched pathways contain many highly m6A modified mRNAs? In addition, they should apply their TE analysis to these enriched pathways. Can they find reduced TE in these highly modified mRNAs that belong to the top KEGG pathways such as BCC and hedgehog signaling?

We apologize for the confusion. We actually used the ranking system when we searched for enriched KEGG pathways, but we agree the way it was written and presented was confusing. We have now clarified this both in the figures and in the text. We expanded our description of the normalization used to emphasize that we have used this ranking of m6A modification over input for this analysis. We have added an analysis to demonstrate that KEGG pathways enriched for the highly modified transcripts tend to be more efficiently translated (new Figure 1—figure supplement 1G). We have also added a new analysis for functional enrichment for genes which are highly m6A modified (top 20%) and have a high translational efficiency (top 20%), as was suggested to in point (1) above. These KEGG pathways in Figure 1G now show the top 8 pathways that contain >5 genes within a category and that surface when we analyze the top 20% of highly modified mRNAs that are also among the top 20% of highly translated mRNAs. As you can see, they are enriched for signaling pathways important for skin lineages, similar to before, but will now be much more convincing and interesting to readers. The statistical parameters suggest the extent of the enrichment. We also add new Figure 1H for readers who don’t necessarily know how these pathways pertain to skin development.

We also greatly thank the reviewers for asking how these pathways for highly methylated, highly translated mRNAs fare upon METTL3 loss. We’ve now separated the scRNA data into significantly down as well as significantly upregulated mRNAs. As shown in new Figure 6A,B, the top 20% of genes that are downregulated in our scRNA seq data include signaling pathway genes (including WNT, SHH and NOTCH), actin regulators and cellular adhesion genes and among these are mRNAs that are heavily modified by m6A. Of particular intrigue, we discovered that many canonical translational initiators are among the 20% most significantly downregulated mRNAs upon *Mettl3* LOF, offering a possible explanation as to why efficiently translated, heavily modified mRNAs tend to be downregulated (Figure 6B).

In our initial version, we did not include any information on the downregulated, heavily modified genes. These new analyses now beautifully tie together the data in Figure 1 with the scRNA seq data and the phenotypic data.

3) The phenotypical analysis should be strengthened by more careful analysis of cell proliferation such as colony formation in addition to EdU and Ki67 staining and cell adhesion analysis. The demonstrated phenotypes also seem similar to Dicer1 KO, when all miRNAs are depleted, including neonatal lethality due to the lack of weight gain, degenerated hair follicles and diminished filiform papillae formation.

The reviewer raises some excellent points. The cKO cells don’t survive in vitro cell culture, as alluded to above, but we now add the colony formation data (new Figure 5—figure supplement 1C). We now show the epidermal flux experiment as part of Figure 6 and with additional data in new Figure 6—figure supplement 2. As NOTCH signaling pathway surfaced among the top 20% of mRNAs modified and top 20% of mRNAs efficiently translated and among the top 20% most significantly downregulated upon METTL3 loss, we also confirm that HES1, the main NOTCH target, is downregulated in the differentiating epidermal cells (new Figure 6D). This is further interesting, as it is consistent with precocious outward flux as well as the increase in cellularity. We corroborate increased cellularity at the ultrastructural level (new Figure 6G). Finally, we show that apoptosis is not increased in the basal layer and thus, cannot account for why proliferation basally is up in the METTL3 LOF pups (new Figure 6—figure supplement 2F). Rather, the data are together, consistent with the increased flux. We did investigate cell adhesion. We see a decrease in P-cad:E-cad ratio, which correlates with a reduction in cell migration and increased departure from the basal layer (new Figure 6—figure supplement 2A). However, even though we saw downregulated *Col17a1, Itgb4* at the mRNA level, we saw hemidesmosomes (Figure 6G). We leave open in the text now the possibility that a decrease in attachment could contribute to enhanced flux.

It is intriguing that to some extent, the phenotype resembles that of mice, whose skin lacks miRNAs. In 2015, Alcaron et al., (2015) used a breast cancer cell line to show that METTL3 methylates pri-miRNAs, marking them for recognition and processing by DGCR8, and that METTL3 depletion results in the global reduction of mature miRNAs and concomitant accumulation of unprocessed pri-miRNAs. While beyond the scope of our current study, our in vivo *Mettl3* LOF coupled with prior Dgrcr8 and Dicer LOF in skin (Yi et al. 2008; 2011) point to an interesting parallel. That said, the skin phenotypes are different in the most fundamental way, and that is that when miRNAs are lost, HFs invaginate, something we don’t see in our mice. Since we don’t analyze miRNAs in our paper, it makes discussion a bit problematic. We have now added some text and references to this effect, in our last few paragraphs of the Discussion section.

4) They mentioned that WNT and SHH pathways surfaced in multiple pathways in their miCLIP data and suggested that m6A might function in hair follicle fate through these pathways. However, the authors did not show how these pathways behaved in WT vs cKO mice. Such analysis will strengthen their claims. Aberrant LEF and SHH signaling was associated with failure to form tight dermal condensates. With this data, they concluded that WNT signaling has been altered by m6A modification. However, there were not mechanistic studies to prove that WNT has been altered. The authors can do qPCR analysis of transcripts that are under WNT regulation to further show altered WNT signaling (e.g. axin) or IHC imaging of β-catenin etc. to future strengthen their conclusion.

This was an excellent point. To this end, we first add data for β-catenin IHC (new Figure 3C), higher magnification of the interface between DP and HF progenitors to show defects in both WNT and SHH signaling in New Figure 3C,D, qPCR results for Lef1 in new Figure 6C as requested and most importantly, a previously not included analysis of the *downregulated* genes from our single cell RNA seq analysis of control and *Mettl3* cKO skin progenitors (new Figure 6A,B). This new analysis shows that among the most significantly downregulated genes are those involved in WNT and SHH signaling. Moreover, a number of these were both highly modified and highly translated in wild-type. Finally, we show that among the mRNAs downregulated are many canonical translation genes, providing further possibilities as to why genes highly modified and efficiently translated are likely down upon *Mettl3* LOF. These data beautifully strengthen our conclusions and tie together the manuscript. We really thank the reviewers for raising this point!

5) They performed single-cell RNA-seq to measure the changed transcriptome and cellular state. However, their cell clustering analysis should be performed by using control and cKO samples together, rather than clustering separately and relying on marker genes to identify each population in control and cKO (Figure 5—figure supplement 1D, F). Are all corresponding cell populations in control and cKO clustered together? If not, they should explore the underlying reasons. The pseudotime analysis in Figure 4B should be strengthened by statistical analysis rather than eyeballing the population/lineage differences between Ctrl and cKO. They also should keep in mind they only have 1 pair of biological samples, which may reduce the robustness of the analysis. For differential gene analysis, they should independently confirm each data (Figure 4C, Figure 5D etc.) by qPCR or bulk RNA-seq from more sample pairs.

For the clustering of cell populations, we did analyze all cells in control and cKO together, so we agree on this point completely with the reviewers. We have now stated this more clearly in the text, as we agree the way we presented the story was confusing. We also now include more details in the Materials and methods section and also show the t-SNE plots with cells from the two conditions projected together in new Figure 5A. We have also performed statistical analysis for the pseudotime. Results are in new Figure 5C. qPCR and IMF verification for the top downregulated genes are substantiated in Figure 3, Figure 4 and Figure 6. Conversely, t-SNE plots, qPCR and IMF verification for upregulated genes of mRNAs highly modified in WT are now added in new Figure 6C, D and new Figure 7—figure supplement 1B-D.

6) Based on ribosome profiling analysis, they suggest that m6A enhances translation. Based on RNA-seq, they suggest that m6A reduces mRNA stability. Globally, these are two opposite effects on mRNA's potential to make protein. It's important to distinguish (1) whether these two opposite effects of m6A are on different mRNAs? (2) if the same mRNAs are under the control of both effects, what is the net effect on protein synthesis? Are they canceled out or one effect is more dominant than the other? How does this explain/correlate with the phenotypes?

Again, these points were incredibly helpful in our revising the manuscript so that others could clearly grasp the significance of our findings. As alluded to above, the comments made us realize that up until the scRNA seq data, we had focused on phenotypes corresponding to highly modified, efficiently translated mRNAs and which belonged to pathways that were suggested to be downregulated. However, in our initial version, we had not delved into the significantly downregulated mRNAs that surfaced upon scRNA seq of *Mettl3* null vs Control transcripts. We’ve now analyzed the most significantly downregulated genes and find that excitingly, these mRNAs encode proteins that are involved in WNT and SHH signaling and actin/ECM/cell adhesion proteins! These data are now presented in a new Figure 6A, B and they add compelling evidence to show that heavily modified, efficiently translated mRNAs involve pathways that are downregulated upon loss of m6A. Additionally, we find that markedly downregulated genes include a number of mRNAs involved in canonical translation (Figure 6B, lower right), further providing a possible explanation for why efficiently translated, highly modified mRNAs might be selectively downregulated upon LOF.

Conversely, the upregulated mRNAs include those which are heavily modified in the wild-type and encode proteins involved in RNA methylation, CAP-independent translation and different aspects of RNA metabolism. So yes, their KEGG profiles and mRNAs are distinct! The data suggest that the upregulated mRNAs are likely involved in (1) rescue mechanisms to boost stalled ribosome machinery and activate alternative pathways for translation; and (2) built-in negative feedback mechanisms for key mRNAs, e.g. the m6A readers (Ythdf1-3) to sense when m6A is low and boost efficiency of reading. Overall, the data beautifully demonstrate that m6A functions in multifaceted ways, which cannot be categorized as simply regulating mRNA translation nor mRNA degradation. The data further show that the functions of down and up regulated mRNAs upon *Mettl3* loss are markedly different—in one case affecting signaling and cell fate decisions and in the other case affecting pathways that become triggered when levels of m6A drop. We now highlight these data in the figures (new Figure 7C, D) and in the text, and add a new Figure 7E model to emphasize these points.

7) The observation of prematurely detached basal cells is interesting. Are they the direct consequence of reduced cell-cell adhesion and/or cell-bm adhesion?

Again, this is an excellent comment. We point out that ECM-receptor interactions are among the top 20% of mRNAs modified heavily and top 20% most efficiently translated. Upon loss of METTL3, Focal adhesion and intercellular adhesion are also categories among the most significantly downregulated in cKO and most heavily m6A modified in Control. This could explain the premature detachment. We provide new data that shows that cell polarity and cell organization are altered (new Figure 4) and that P-cad:E-cad is down in the cKO (Figure 6—figure supplement 2A), expected for a decrease in migration/actin dynamics normally seen in basal cells. Our new data point to NOTCH pathway genes as being highly m6A modified and efficiently translated and that NOTCH signaling is downregulated in the cKO epidermal progenitors as they exit the basal layer (new Figure 6D). Additionally, ECM-receptor interaction and actin regulators are also among the most significantly downregulated upon LOF. These data presented in new Figure 6B are supportive of the reviewers’ hypothesis. However, electron microscopy showed that although the epidermis is highly disordered and show increased cellularity and signs of basal features suprabasally, overall hemidesmosomes and intercellular junctions are largely intact, even at P0 (new Figure 6G). This is corroborated by TEWL, showing intact barrier function in newborn mice. Our data together point to the notion that precocious departure of basal progenitors is occurring and is reflected in basal features lingering suprabasally, but these defects are subtle. We attribute this to a remarkable ability of the epidermal progenitors to activate compensatory mechanisms that work for a while until eventually, the deleterious effects of m6A loss cannot be overcome. We thank the reviewers for encouraging us to explore this avenue in depth, and although our answers are not simple, they do provide valuable insights for future explorations.

8) The reviewers were not sure about the point of OPP experiments. OPP label may be too low resolution to show any changes in Mettl3 cKO.

We agree and felt that in the absence of ribosomal profiling, the data is best omitted, which we’ve done. However, our new data in Figure 6 show that canonical translational regulators are significantly downregulated upon METTL3 LOF as are signaling pathways whose mRNAs include among the most highly modified and most efficiently translated (new Figure 6B). Conversely, CAP-independent translational regulators are upregulated as are other Rna processing factors that suggest alternative pathways for translation (new Figure 7C-E). Finally, we also see signs of rescue mechanisms. First is Myc, known to function in enhanced translation (new Figure 7—figure supplement 1B-D). Second is YTHDF1, YTHDF2 and YTHDF3, all of which we show are highly methylated by m6A in WT progenitors and highly upregulated upon m6A loss. As these are key m6A readers—affecting translation (1 and 3) and RNA degradation (2) they too are suggestive of a means to boost m6A recognition in the face of m6A deficiency. These arguments are stronger and more interesting than the OPP data.

9) Figure 3E-F: it is clear that in the grafted skin Mettl3 cKO HFs underwent HF degeneration in accompany with sebocyte differentiation. However, it is not clear if this fate switch only occurred in Mettl3 cKO HFs upon wounding (grafting in this case). Oil red O staining with P6 ungrafted skin sections will partially address this question. If this fate switch only happens upon wounding, authors should discuss the effects of m6A loss in the differential circumstance as well as speculate the effect of wounding in m6A-mediated regulation.

Excellent point! We have performed the Oil red O staining on P6 ungrafted skin (new Figure 3E) and added discussion about the potential influences caused by wounding.

10) Subsection “Conditional ablation of Mettl3 in epidermal progenitors results in a marked defect in HF morphogenesis”: "the engrafted cKO epidermis was hyperthickened, a feature which had also been evident in ungrafted cKO epidermis." To this reviewer, the hyperthicken epidermis could NOT be appreciated in ungrafted cKO epidermis based on the H&E staining shown in Figure 2G. The quantification in the thickness of ungrafted and grafted epidermis will be essential. In line with point #2, this hyperthicken epidermis in grafted cKO skin might be caused by wounding if hyperthickness is not detected in ungrafted Mettl3 cKO epidermis.

We agree and feel that the point is best omitted from the text, even though we did quantify and confirmed the thickness. Our EM data showing increased cellularity is clear and supported by quantifications at the IMF level (new Figure 6G; Figure 6—figure supplement 2B).

11) Figure 5E: Authors stated that MYC expression was elevated within Mettl3 cKO epidermal progenitors (subsection “A cohort of highly m6A-decorated mRNAs whose levels rise upon removal of their modification”). It is not clear to this reviewer how the quantification of MYC expression was done. Which cell markers did authors use to define "epidermal progenitors" (e.g. integrin α6 or K14…)? Did authors also include P-cad+ WNY-hi, WNT-lo HF cells, and epidermal suprabasal cells? Please clarify this by adding information in the figure legend as well as Materials and methods section.

We now clarify this. We identified populations from our scRNA seq data according to known marker gene expressions (Figure 5B; now color coded as in the t-SNE plots in Figure 5A). For our substantiating qPCR, we used basal skin progenitors FACS-purified from whole E16.5 skins as shown in Figure 5—figure supplement 1E. We now state this in the text and methods. For MYC, we quantified the t-SNE data also by IMF and there we focus on basal progenitors at both E17.5 and P0. We moved the MYC data to the supplement as it is confusing to most readers not in the skin field. The Watt group showed that MYC functions in terminal differentiation of the epidermis while others have shown that *Myc* can act as a WNT target gene in e.g. colon. In the APC LOF skin mouse, *Myc* is actually downregulated even though WNT signaling is upregulated. Thus, although our data are consistent with the antagonistic effects of WNT signaling and MYC in the skin, it is not an essential point to our work here.

12) Figure 3—figure supplement 1: It is interesting to see high levels of PCAD expression in the basal layer of grafted Mettl3 cKO epidermis. Did authors also find this striking pattern in ungrafted Mettl3 cKO epidermis (e.g. P6 ungrafted skin)? What could be the explanation of this significant upregulation of P-Cadherin in the Mettl3 cKO epidermal basal cells, e.g. alteration in cell junction? Confusion in fate specification? It will be helpful if authors can check on ungrafted skin and speculate the possibilities.

First, we have selected a more representative image for the PCAD staining in the grafted skin (Figure 3—figure supplement 1). Second, we have examined the expression levels of PCAD and ECAD at E16.5 closely. Results are included in Figure 6—figure supplement 2A. Our data show clearly that P-cadherin:E-cadherin ratios shift in the *Mettl3* cKO. We also provide data that there are some gaps, visible by EM in the adherens junctions of the cKO vs WT, but these are subtle (Figure 6G). We think that these differences, coupled with the marked downregulation of actin regulators, polarity genes and adhesion genes in the cKO (Figure 6B, Figure 4B) are all suggestive that there are subtle defects here. We show the data and discuss this possibility.